# HYPERBOLIC NEURAL NETWORKS++

**Ryohei Shimizu**[1], **Yusuke Mukuta**[1,2], **Tatsuya Harada**[1,2]
[1]The University of Tokyo
[2]RIKEN AIP
{shimizu, mukuta, harada}@mi.t.u-tokyo.ac.jp

## ABSTRACT

Hyperbolic spaces, which have the capacity to embed tree structures without distortion owing to their exponential volume growth, have recently been applied to machine learning to better capture the hierarchical nature of data. In this study, we generalize the fundamental components of neural networks in a single hyperbolic geometry model, namely, the Poincaré ball model. This novel methodology constructs a multinomial logistic regression, fully-connected layers, convolutional layers, and attention mechanisms under a unified mathematical interpretation, without increasing the parameters. Experiments show the superior parameter efficiency of our methods compared to conventional hyperbolic components, and stability and outperformance over their Euclidean counterparts.

## 1 INTRODUCTION

Shifting the arithmetic stage of a neural network to a non-Euclidean geometry such as a hyperbolic space is a promising way to find more suitable geometric structures for representing or processing data. Owing to its exponential growth in volume with respect to its radius (Krioukov et al., 2009; 2010), a hyperbolic space has the capacity to continuously embed tree structures with arbitrarily low distortion (Krioukov et al., 2010; Sala et al., 2018). It has been directly utilized, for instance, to visualize large taxonomic graphs (Lamping et al., 1995), to embed scale-free graphs (Blasius et al., 2018), or to learn hierarchical lexical entailments (Nickel & Kiela, 2017). Compared to the Euclidean space, a hyperbolic space shows a higher embedding accuracy under fewer dimensions in such cases.

Because a wide variety of real-world data encompasses some type of latent hierarchical structures (Katayama & Maina, 2015; Newman, 2005; Lin & Tegmark, 2017; Krioukov et al., 2010), it has been empirically proven that a hyperbolic space is able to capture such intrinsic features through representation learning (Krioukov et al., 2010; Ganea et al., 2018b; Nickel & Kiela, 2018; Tifrea et al., 2019; Law et al., 2019; Balazevic et al., 2019; Gu et al., 2019). Motivated by such expressive characteristics, various machine learning methods, including support vector machines (Cho et al., 2019) and neural networks (Ganea et al., 2018a; Gulcehre et al., 2018; Micic & Chu, 2018; Chami et al., 2019) have derived the analogous benefits from the introduction of a hyperbolic space, aiming to improve the performance on advanced tasks beyond just representing data.

One of the pioneers in this area is Hyperbolic Neural Networks (HNNs), which introduced an easy-to-interpret and highly analytical coordinate system of hyperbolic spaces, namely, the Poincaré ball model, with a corresponding gyrovector space to smoothly connect the fundamental functions common to neural networks into valid functions in a hyperbolic geometry (Ganea et al., 2018a). Built upon the solid foundation of HNNs, the essential components for neural networks covering the multinomial logistic regression (MLR), fully-connected (FC) layers, and Recurrent Neural Networks have been realized. In addition to the formalism, the methods for graphs (Liu et al., 2019), sequential classification (Micic & Chu, 2018), or Variational Autoencoders (Nagano et al., 2019; Mathieu et al., 2019; Ovinnikov, 2019; Skopek et al., 2020) are further constructed. Such studies have applied the Poincaré ball model as a natural and viable option in the area of deep learning.

Despite such progress, however, there still remain some unsolved problems and uncovered regions. In terms of the network architectures, the current formulation of hyperbolic MLR (Ganea et al., 2018a) requires almost twice the number of parameters compared to its Euclidean counterpart. This makes both the training and inference costly in cases in which numerous embedded entities should be

classified or where large hidden dimensions are employed, such as in natural language processing. The lack of convolutional layers must also be mentioned, because their application is now ubiquitous and is no longer limited to the field of computer vision.

For the individual functions that are commonly used in machine learning, the split and concatenation of vectors have yet to be realized in a hyperbolic space in a manner that can fully exploit such space and allow sub-vectors to achieve a commutative property. Additionally, although several types of closed-form centroids in a hyperbolic space have been proposed, their geometric relationships have not yet been analyzed enough. Because a centroid operation has been utilized in many recent attention-based architectures, the theoretical background for which type of hyperbolic centroid should be used would be required in order to properly convert such operations into the hyperbolic geometry.

Based on the previous analysis, we reconsider the flow of several extensions to bridge Euclidean operations into hyperbolic operations and construct alternative or novel methods on the Poincaré ball model. Specifically, the main contributions of this paper are summarized as follows:

1. We reformulate a hyperbolic MLR to reduce the number of parameters to the same level as a Euclidean version while maintaining the same range of representational properties.

2. We further exploit the knowledge of 1 as a replacement of an affine transformation and propose a novel generalization of the FC layers that can more properly make use of the hyperbolic nature compared with a previous research (Ganea et al., 2018a).

3. We generalize the split and concatenation of coordinates to the Poincaré ball model by setting the invariance of the expected value of the vector norm as a criterion.

4. By combining 2 and 3, we further define a novel generalization scheme of arbitrary dimensional convolutional layers in the Poincaré ball model.

5. We prove the equivalence of the hyperbolic centroids defined in three different hyperbolic geometry models, and expand the condition of non-negative weights to entire real values. Moreover, integrating this finding and previous contributions 1, 2, and 3, we give a theoretical insight into hyperbolic attention mechanisms realized in the Poincaré ball model.

We experimentally demonstrate the effectiveness of our methods over existing HNNs and Euclidean equivalents based on a performance test of MLR functions and experiments with Set Transformer (Lee et al., 2019) and convolutional sequence to sequence modeling (Gehring et al., 2017).[1]

## 2 HYPERBOLIC GEOMETRY

**Riemannian geometry.** An $n$-dimensional manifold $\mathcal{M}$ is an $n$-dimensional topological space that can be linearly approximated to an $n$-dimensional real space at any point $\boldsymbol{x} \in \mathcal{M}$, and each local linear space is called a tangent space $\mathcal{T}_{\boldsymbol{x}}\mathcal{M}$. A Riemannian manifold is a pairing of a differentiable manifold and a metric tensor field $\mathfrak{g}$ as a function of each point $\boldsymbol{x}$, which is expressed as $(\mathcal{M}, \mathfrak{g})$. Here, $\mathfrak{g}$ defines an inner product on each tangent space such that $^{\forall}\boldsymbol{u}, \boldsymbol{v} \in \mathcal{T}_{\boldsymbol{x}}\mathcal{M}, \langle \boldsymbol{u}, \boldsymbol{v} \rangle_{\boldsymbol{x}} = \boldsymbol{u}^{\top} \mathfrak{g}_{\boldsymbol{x}} \boldsymbol{v}$, where $\mathfrak{g}_{\boldsymbol{x}}$ is a positive definite symmetric matrix defined on $\mathcal{T}_{\boldsymbol{x}}\mathcal{M}$. The norm of a tangent vector derived from the inner product is defined as $\|\boldsymbol{v}\|_{\boldsymbol{x}} = \sqrt{|\langle \boldsymbol{v}, \boldsymbol{v} \rangle_{\boldsymbol{x}}|}$. A metric tensor $\mathfrak{g}_{\boldsymbol{x}}$ provides local information regarding the angle and length of the tangent vectors in $\mathcal{T}_{\boldsymbol{x}}\mathcal{M}$, which induces the global length of the curves on $\mathcal{M}$ through an integration. The shortest path connecting two arbitrary points on $\mathcal{M}$ at a constant speed is called a geodesic, the length of which becomes the distance. Along a geodesic where one of the endpoints is $\boldsymbol{x}$, the function projecting a tangent vector $\boldsymbol{v} \in \mathcal{T}_{\boldsymbol{x}}\mathcal{M}$ as an initial velocity vector onto $\mathcal{M}$ is denoted as an exponential map $\exp_{\boldsymbol{x}}$, and its inverse function is called a logarithmic map $\log_{\boldsymbol{x}}$. In addition, the concept of parallel transport $P_{\boldsymbol{x} \to \boldsymbol{y}} : \mathcal{T}_{\boldsymbol{x}}\mathcal{M} \to \mathcal{T}_{\boldsymbol{y}}\mathcal{M}$ is generalized to the specially conditioned unique linear isometry between two tangent spaces. For more details, please refer to Spivak (1979); Petersen et al. (2006); Andrews & Hopper (2010).

Note that, in this study, we equate $\mathfrak{g}$ with $\mathfrak{g}_{\boldsymbol{x}}$ if $\mathfrak{g}_{\boldsymbol{x}}$ is constant, and denote the Euclidean inner product, norm, and unit vector for any real vector $\boldsymbol{u}, \boldsymbol{v} \in \mathbb{R}^n$ as $\langle \boldsymbol{u}, \boldsymbol{v} \rangle$, $\|\boldsymbol{v}\|$, and $[\boldsymbol{v}] = \boldsymbol{v}/\|\boldsymbol{v}\|$, respectively.

**Hyperbolic space.** A hyperbolic space is a Riemannian manifold with a constant negative curvature, the coordinates of which can be represented in several isometric models. The most basic model is an

---

[1]The code is available at https://github.com/mil-tokyo/hyperbolic_nn_plusplus.

$n$-dimensional hyperboloid model, which is a hypersurface $\mathbb{H}_c^n$ in an $(n+1)$-dimensional Minkowski space $\mathbb{R}_1^{n+1}$ composed of one time-like axis and $n$ space-like axes. The manifolds of Poincaré ball model $\mathbb{B}_c^n$ and Beltrami-Klein model $\mathbb{K}_c^n$ are the projections of the hyperboloid model onto the different $n$-dimensional space-like hyperplanes, as depicted in Figure 1. For their mathematical definitions and the isometric isomorphism between their coordinates, see Appendix A.

**Poincaré ball model.** The $n$-dimensional Poincaré ball model of a constant negative curvature $-c$ is defined by $(\mathbb{B}_c^n, \mathfrak{g}^c)$, where $\mathbb{B}_c^n = \{\boldsymbol{x} \in \mathbb{R}^n \mid c\|\boldsymbol{x}\|^2 < 1\}$ and $\mathfrak{g}_{\boldsymbol{x}}^c = (\lambda_{\boldsymbol{x}}^c)^2 \boldsymbol{I}_n$. Here, $\mathbb{B}_c^n$ is an open ball of radius $c^{-\frac{1}{2}}$, and $\lambda_{\boldsymbol{x}}^c = 2(1 - c\|\boldsymbol{x}\|^2)^{-1}$ is a conformal factor, which induces the inner product $\langle \boldsymbol{u}, \boldsymbol{v} \rangle_{\boldsymbol{x}}^c = (\lambda_{\boldsymbol{x}}^c)^2 \langle \boldsymbol{u}, \boldsymbol{v} \rangle$ and norm $\|\boldsymbol{v}\|_{\boldsymbol{x}}^c = \lambda_{\boldsymbol{x}}^c \|\boldsymbol{v}\|$ for $\boldsymbol{u}, \boldsymbol{v} \in \mathcal{T}_{\boldsymbol{x}} \mathbb{B}_c^n$. The exponential, logarithmic maps and parallel transport are denoted as $\exp_{\boldsymbol{x}}^c$, $\log_{\boldsymbol{x}}^c$ and $P_{\boldsymbol{x} \to \boldsymbol{y}}^c$, respectively, as shown in Appendix C.

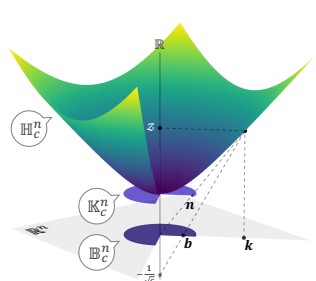

To operate the coordinates as vector-like mathematical objects, the Möbius gyrovector space provides an algebra that treats them as gyrovectors, equipped with various operations including the generalized vector addition, that is, a noncommutative and non-associative binary operation called the Möbius addition $\oplus_c$ (Ungar, 2009). $\lim_{c \to 0} \oplus_c$ converges to $+$ in connection with a Euclidean geometry, the curvature of which is zero. For more details, see Appendix B.

Figure 1: Geometric relationship between $\mathbb{H}_c^n$, $\mathbb{B}_c^n$ and $\mathbb{K}_c^n$ depicted in $\mathbb{R}_1^{n+1}$.

**Poincaré hyperplane.** As a specific generalization of a hyperplane into Riemannian geometry, Ganea et al. (2018a) derived a Poincaré hyperplane $\tilde{H}_{\boldsymbol{a},\boldsymbol{p}}^c$, which is the set of all geodesics containing an arbitrary point $\boldsymbol{p} \in \mathbb{B}_c^n$ and orthogonal to an arbitrary tangent vector $\boldsymbol{a} \in \mathcal{T}_{\boldsymbol{p}} \mathbb{B}_c^n$, based on the Möbius gyrovector space. As shown in Appendix C.2, they also extended the distance $d_c$ between two points in $\mathbb{B}_c^n$ into the distance from a point in $\mathbb{B}_c^n$ to a Poincaré hyperplane in a closed form expression.

## 3 HYPERBOLIC NEURAL NETWORKS++

Aiming to overcome the difficulties discussed in Section 1, we build a novel scheme of hyperbolic neural networks in the Poincaré ball model. The core concept is re-generalization of $\langle \boldsymbol{a}, \boldsymbol{x} \rangle - b$ type equations with no increase in the number of parameters, which has the potential to replace any affine transformation based on the same mathematical principle. Specifically, this section starts from the reformulation of the hyperbolic MLR, from which the variants to the FC, convolutional, and multi-head attention layers are derived. Several other modifications are also proposed to support neural networks with broad architectures.

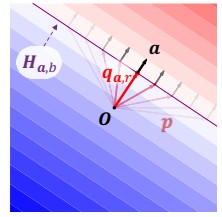 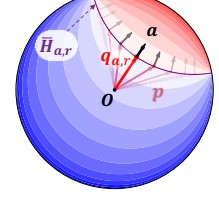

(a) Reformulation in $\mathbb{R}^n$    (b) Generalization to $\mathbb{B}_c^n$

Figure 2: Whichever pair of $\boldsymbol{a}$ and $\boldsymbol{p}$ is chosen, it determines the same discriminative hyperplane. Considering one bias point $\boldsymbol{q}_{\boldsymbol{a},r}$ per one discriminative hyperplane solves this over-parameterization.

### 3.1 UNIDIRECTIONAL REPARAMETERIZATION OF HYPERBOLIC MLR LAYER

Given an input $\boldsymbol{x} \in \mathbb{R}^n$, MLR is an operation used to predict the probabilities of all target outcomes $k \in \{1, 2, ..., K\}$ for the objective variable $y$ as a log-linear model and is described as follows:

$$p(y = k \mid \boldsymbol{x}) \propto \exp\left(v_k(\boldsymbol{x})\right), \quad \text{where } v_k(\boldsymbol{x}) = \langle \boldsymbol{a}_k, \boldsymbol{x} \rangle - b_k, \ \boldsymbol{a}_k \in \mathbb{R}^n, \ b_k \in \mathbb{R}. \quad (1)$$

**Circumvention of the double vectorization.** To generalize the linear function $v_k$ to the Poincaré ball model, Ganea et al. (2018a) first re-parameterized the scalar term $b_k$ as a vector $\boldsymbol{p}_k \in \mathbb{R}^n$ in the form $\langle \boldsymbol{a}_k, \boldsymbol{x} \rangle - b_k = \langle \boldsymbol{a}_k, -\boldsymbol{p}_k + \boldsymbol{x} \rangle$, where $b_k = \langle \boldsymbol{a}_k, \boldsymbol{p}_k \rangle$, and then discussed the properties which must be satisfied when such vectors become Möbius gyrovectors. However, this causes an undesirable increase in the parameters from $n + 1$ to $2n$ in each class $k$. As illustrated in Figure 2 (a), this reformulation is redundant from the viewpoint that there exist countless choices of $\boldsymbol{p}_k$ to determine the same discriminative hyperplane $H_{\boldsymbol{a}_k, b_k} = \{\boldsymbol{x} \in \mathbb{R}^n \mid \langle \boldsymbol{a}_k, \boldsymbol{x} \rangle - b_k = 0\}$. Because the

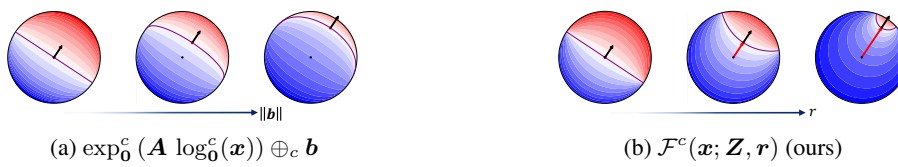

(a) $\exp_{\mathbf{0}}^c\left(\boldsymbol{A}\,\log_{\mathbf{0}}^c(\boldsymbol{x})\right) \oplus_c \boldsymbol{b}$         (b) $\mathcal{F}^c(\boldsymbol{x}; \boldsymbol{Z}, \boldsymbol{r})$ (ours)

Figure 3: Comparison of FC layers in input spaces $\mathbb{B}_c^n$. The values at a certain dimension of output spaces are illustrated as contour plots. Black arrows depict the orientation parameters, and they are fixed for the comparison. Their orthogonal curves show discriminative hyperplanes where the values are zeros. As a bias parameter $\boldsymbol{b}$ or $r_k$ changes, the outline of the contour landscape in (a) remains unchanged, whereas in (b) the focused regions are dynamically squeezed according to the geodesics.

key of this step is to replace all variables with vectors attributed to the same manifold, we introduce another scalar parameter $r_k \in \mathbb{R}$ instead, which makes the bias vector $\boldsymbol{q}_{\boldsymbol{a}_k,r_k}$ parallel to $\boldsymbol{a}_k$:

$$\langle \boldsymbol{a}_k, \boldsymbol{x} \rangle - b_k = \langle \boldsymbol{a}_k, -\boldsymbol{q}_{\boldsymbol{a}_k,r_k} + \boldsymbol{x} \rangle, \quad \text{where} \quad \boldsymbol{q}_{\boldsymbol{a}_k,r_k} = r_k[\boldsymbol{a}_k] \quad s.t. \quad b_k = r_k\|\boldsymbol{a}_k\|. \tag{2}$$

One possible realization of $\boldsymbol{p}_k$ is adopted to reduce the previously mentioned redundancies without a loss of generality or representational properties compared to the original affine transformation, and induces another notation: $\bar{H}_{\boldsymbol{a}_k,r_k} := \{\boldsymbol{x} \in \mathbb{R}^n \mid \langle \boldsymbol{a}_k, -\boldsymbol{q}_{\boldsymbol{a}_k,r_k} + \boldsymbol{x} \rangle = 0\} = H_{\boldsymbol{a}_k,r_k\|\boldsymbol{a}_k\|}$. Based on distance $d$ from a point to a hyperplane, Equation 2 can be rewritten as with Lebanon & Lafferty (2004) in the following form: $\langle \boldsymbol{a}_k, -\boldsymbol{q}_{\boldsymbol{a}_k,r_k} + \boldsymbol{x} \rangle = \text{sign}(\langle \boldsymbol{a}_k, -\boldsymbol{q}_{\boldsymbol{a}_k,r_k} + \boldsymbol{x} \rangle)\, d(\boldsymbol{x}, \bar{H}_{\boldsymbol{a}_k,r_k})\|\boldsymbol{a}_k\|$, which decomposes the inner product into the product of the norm of an orientation vector $\boldsymbol{a}_k$ and the signed distance between an input vector $\boldsymbol{x} \in \mathbb{R}^n$ and the hyperplane $\bar{H}_{\boldsymbol{a}_k,r_k}$.

**Unidirectional Poincaré MLR.** Based on the observation that $\boldsymbol{q}_{\boldsymbol{a}_k,r_k}$ starts from the origin and the concept of Poincaré hyperplanes, we can now generalize $v_k$ for $\boldsymbol{x}$, $\boldsymbol{q}_{\boldsymbol{a}_k,r_k} \in \mathbb{B}_c^n$ and $\boldsymbol{a}_k \in \mathcal{T}_{\boldsymbol{q}_{\boldsymbol{a}_k,r_k}}\mathbb{B}_c^n$:

$$v_k(\boldsymbol{x}) = \text{sign}(\langle \boldsymbol{a}_k, \ominus_c \boldsymbol{q}_{\boldsymbol{a}_k,r_k} \oplus_c \boldsymbol{x} \rangle)\, d_c\left(\boldsymbol{x}, \bar{H}_{\boldsymbol{a}_k,r_k}^c\right)\|\boldsymbol{a}_k\|_{\boldsymbol{q}_{\boldsymbol{a}_k,r_k}}^c, \tag{3}$$

$$\text{where} \quad \boldsymbol{q}_{\boldsymbol{a}_k,r_k} = \exp_{\mathbf{0}}^c(r_k[\boldsymbol{a}_k]), \quad \bar{H}_{\boldsymbol{a}_k,r_k}^c := \{\boldsymbol{x} \in \mathbb{B}_c^n \mid \langle \boldsymbol{a}_k, \ominus_c \boldsymbol{q}_{\boldsymbol{a}_k,r_k} \oplus_c \boldsymbol{x} \rangle = 0\}, \tag{4}$$

which are shown in Figure 2 (b). Importantly, the circular reference between $\boldsymbol{a}_k \in \mathcal{T}_{\boldsymbol{q}_{\boldsymbol{a}_k,r_k}}\mathbb{B}_c^n$ and $\boldsymbol{q}_{\boldsymbol{a}_k,r_k}$ can be unraveled by considering the tangent vector at the origin, $\boldsymbol{z}_k \in \mathcal{T}_{\mathbf{0}}\mathbb{B}_c^n$, from which $\boldsymbol{a}_k$ is parallel transported by $P_{\boldsymbol{x}\to\boldsymbol{y}}^c : \mathcal{T}_{\boldsymbol{x}}\mathbb{B}_c^n \to \mathcal{T}_{\boldsymbol{y}}\mathbb{B}_c^n$ described in Appendix C.3 as follows:

$$\boldsymbol{a}_k = P_{\mathbf{0}\to\boldsymbol{q}_{\boldsymbol{a}_k,r_k}}^c(\boldsymbol{z}_k) = \text{sech}^2\left(\sqrt{c}\,r_k\right)\boldsymbol{z}_k, \quad \boldsymbol{q}_{\boldsymbol{a}_k,r_k} = \exp_{\mathbf{0}}^c(r_k[\boldsymbol{z}_k]) = \boldsymbol{q}_{\boldsymbol{z}_k,r_k}. \tag{5}$$

Combining Equations 3, 5, and 23, we conclude the derivation of the unidirectional re-generalization of MLR, the parameters of which are $r_k \in \mathbb{R}$ and $\boldsymbol{z}_k \in \mathcal{T}_{\mathbf{0}}\mathbb{B}_c^n = \mathbb{R}^n$ for each class $k$:

$$v_k(\boldsymbol{x}) = 2\,c^{-\frac{1}{2}}\|\boldsymbol{z}_k\| \sinh^{-1}\left(\lambda_{\boldsymbol{x}}^c\langle \sqrt{c}\,\boldsymbol{x}, [\boldsymbol{z}_k]\rangle \cosh\left(2\sqrt{c}\,r_k\right) - (\lambda_{\boldsymbol{x}}^c - 1)\sinh\left(2\sqrt{c}\,r_k\right)\right). \tag{6}$$

For more detailed deformation, see Appendix D.1. Note that we recover the form of the standard Euclidean MLR in $\lim_{c\to 0} v_k(\boldsymbol{x}) = 4(\langle \boldsymbol{a}_k, \boldsymbol{x} \rangle - b_k)$, which is proven in Appendix D.2.

## 3.2 REFORMULATING FC LAYERS TO PROPERLY EXPLOIT THE HYPERBOLIC PROPERTIES

We next discuss the FC layers, described as a simple affine transformation $\boldsymbol{y} = \boldsymbol{A}\boldsymbol{x} - \boldsymbol{b}$, in an element-wise manner with respect to the output space as $y_k = \langle \boldsymbol{a}_k, \boldsymbol{x} \rangle - b_k$, where $\boldsymbol{x}, \boldsymbol{a}_k \in \mathbb{R}^n$ and $b_k \in \mathbb{R}$. This can be interpreted as an operation that linearly transforms the input $\boldsymbol{x}$ and treats the output score $y_k$ as the coordinate value at, or the signed distance from the hyperplane containing the origin and orthogonal to, the $k$-th axis of the output space $\mathbb{R}^m$. Therefore, combining them with a generalized linear transformation, as described in Section 3.1, we can now generalize the FC layers:

**Poincaré FC layer.** Given an input $\boldsymbol{x} \in \mathbb{B}_c^n$, with the generalized linear transformation $v_k$ in Equation 6 and the parameters composed of $\boldsymbol{Z} = \{\boldsymbol{z}_k \in \mathcal{T}_{\mathbf{0}}\mathbb{B}_c^n = \mathbb{R}^n\}_{k=1}^m$, which is a generalization of $\boldsymbol{A}$ and $\boldsymbol{r} = \{r_k \in \mathbb{R}\}_{k=1}^m$ representing the bias terms, the Poincaré FC layer outputs the following:

$$\boldsymbol{y} = \mathcal{F}^c(\boldsymbol{x}; \boldsymbol{Z}, \boldsymbol{r}) := \boldsymbol{w}(1 + \sqrt{1 + c\|\boldsymbol{w}\|^2})^{-1}, \quad \text{where} \quad \boldsymbol{w} := (c^{-\frac{1}{2}}\sinh\left(\sqrt{c}\,v_k(\boldsymbol{x})\right))_{k=1}^m. \tag{7}$$

It can be proven that the signed distance from $\boldsymbol{y}$ to each Poincaré hyperplane containing the origin, and orthogonal to the $k$-th axis, equals $v_k(\boldsymbol{x})$, as shown in Appendix D.3, satisfying the aforementioned properties. We also recover a FC layer in $\lim_{c\to 0} y_k = 4\left(\langle \boldsymbol{a}_k, \boldsymbol{x} \rangle - r_k\|\boldsymbol{a}_k\|\right)$.

**Comparison with a previous method.** Ganea et al. (2018a) proposed a hyperbolic FC layer operating a matrix-vector multiplication in a tangent space and adding a bias through the following Möbius addition: $\boldsymbol{y} = \exp_{\boldsymbol{0}}^c \left( \boldsymbol{A} \log_{\boldsymbol{0}}^c(\boldsymbol{x}) \right) \oplus_c \boldsymbol{b}$, which indicates that the discriminative hyperplane determined in $\mathcal{T}_{\boldsymbol{0}} \mathbb{B}_c^m$ is projected back to $\mathbb{B}_c^m$ by the exponential map at the origin. However, such a surface is no longer a Poincaré hyperplane, except for $\boldsymbol{b} = \boldsymbol{0}$. Moreover, the basic shape of the contour surfaces in the output space $\mathbb{B}_c^m$ is determined only by the orientation of each row vector $\boldsymbol{a}_k$ in $\boldsymbol{A}$, whereas their norms and a bias term $\boldsymbol{b}$ contribute to the total scale and shift. Conversely, the parameters in our method cooperate to realize more various contour surfaces. Notably, discriminative hyperplanes become Poincaré hyperplanes, *i.e.*, the set of all geodesics orthogonal to the orientation $\boldsymbol{z}_k$ and containing a point $\exp_{\boldsymbol{0}}^c(r_k[\boldsymbol{z}_k])$. As shown in Figure 3, the input space $\mathbb{B}_c^n$ is separated in a more meaningful manner as a hyperbolic space for each dimension of the output space $\mathbb{B}_c^m$.

### 3.3 REGULARIZING SPLIT AND CONCATENATION

Split and concatenation are essential operations for realizing small process branches in parallel or combining feature vectors. However, in the Poincaré ball model, merely splitting the coordinates lowers the norms of the output gyrovectors and limits the representational power, and concatenating them is invalid because the norm of the output can easily exceed the domain of the ball. One simple solution is to conduct an operation in the tangent space. The aforementioned problem regarding a split operation, however, remains. Moreover, as the number of inputs to be concatenated increases, the output gyrovector approaches the boundary of the ball even if the norm of each input is adequately small. The norm of the gyrovector is crucial in the Poincaré ball model owing to its metric. Therefore, reflecting the orientation of inputs while preserving the scale of the norm is considered to be desirable.

**Generalization criterion.** In Euclidean neural networks, keeping the variance of feature vectors constant is an essential criterion (He et al., 2015). As an analogy, keeping the expected values of the norms constant is a worthy criterion in the Poincaré ball because the norm of any Möbius gyrovector is upper-bounded by the ball radius and the variance of the coordinates cannot necessarily remain intact when the dimensions in each layer vary. Such a replacement of the statistic invariance target from each coordinate to the norm is also suggested by Becigneul & Ganea (2019). To satisfy this criterion, we propose the following generalization scheme with a scalar coefficient $\beta_n = \mathrm{B}(\frac{n}{2}, \frac{1}{2})$, where B indicates a beta function.

**Poincaré $\beta$-split.** First, the input $\boldsymbol{x} \in \mathbb{B}_c^n$ is split in the tangent space with integers $s.t.\ \sum_{i=1}^N n_i = n$: $\boldsymbol{x} \mapsto \boldsymbol{v} = \log_{\boldsymbol{0}}^c(\boldsymbol{x}) = (\boldsymbol{v}_1^\top \in \mathbb{R}^{n_1}, \ldots, \boldsymbol{v}_N^\top \in \mathbb{R}^{n_N})^\top$. Each split tangent vector is then properly scaled and projected back to the Poincaré ball as follows: $\boldsymbol{v}_i \mapsto \boldsymbol{y}_i = \exp_{\boldsymbol{0}}^c \left( \beta_{n_i} \beta_n^{-1} \boldsymbol{v}_i \right)$.

**Poincaré $\beta$-concatenation.** Likewise, the inputs $\{ \boldsymbol{x}_i \in \mathbb{B}_c^{n_i} \}_{i=1}^N$ are first properly scaled and concatenated in the tangent space, and then projected back to the Poincaré ball in the following manner: $\boldsymbol{x}_i \mapsto \boldsymbol{v}_i = \log_{\boldsymbol{0}}^c(\boldsymbol{x}_i) \in \mathcal{T}_{\boldsymbol{0}} \mathbb{B}_c^{n_i}$, $\boldsymbol{v} := (\beta_n \beta_{n_1}^{-1} \boldsymbol{v}_1^\top, \ldots, \beta_n \beta_{n_N}^{-1} \boldsymbol{v}_N^\top)^\top \mapsto \boldsymbol{y} = \exp_{\boldsymbol{0}}^c(\boldsymbol{v})$.

We prove the previously mentioned properties under a certain assumption in Appendix D.4. One can also confirm that the Poincaré $\beta$-concatenation is the inverse function of the Poincaré $\beta$-split.

**Discussion about the concatenation.** Ganea et al. (2018a) generalized a vector concatenation under the premise that the output must be followed by an FC layer, but such an assumption possibly limits its usage. Furthermore, it requires Möbius additions $N-1$ times sequentially due to the noncommutative and non-associative properties of the Möbius addition, which incurs a heavy computational cost and an unbalanced priority in each input gyrovector. Alternatively, our method with a pair of exponential and logarithmic maps has a lower computational cost regardless of $N$ and treats every input fairly.

### 3.4 ARBITRARY DIMENSIONAL CONVOLUTIONAL LAYER

The activation of $D$-dimensional convolutional layers with kernel sizes of $\{K_i\}_{i=1}^D$ is generally described as an affine transformation $y_k = \langle \boldsymbol{a}_k, \boldsymbol{x} \rangle - b_k$ for each channel $k$, where $\boldsymbol{x} \in \mathbb{R}^{nK}$ is an input vector per pixel, and is a concatenation of $K = \prod_i K_i$ feature vectors contained in a receptive field of the kernel. This notation also includes a dilated operation. It is now natural to generalize the convolutional layers with Poincaré $\beta$-concatenation and a Poincaré FC layer.

**Poincaré convolutional layer.** At each pixel in the given feature map, the gyrovectors $\{\boldsymbol{x}_s \in \mathbb{B}_c^n\}_{s=1}^K$ contained in a receptive field of the kernel are concatenated into a single gyrovector $\boldsymbol{x} \in \mathbb{B}_c^{nK}$ in the manner proposed in Section 3.3, which is then operated in the same way as a Poincaré FC layer.

### 3.5 ANALYSIS OF HYPERBOLIC ATTENTION MECHANISMS IN THE POINCARÉ BALL MODEL

As preparation for constructing hyperbolic attention mechanisms, it is necessary to theoretically consider the midpoint operation of multiple coordinates in a hyperbolic space. For the Poincaré ball model and Beltrami-Klein model, Ungar (2009) proposed the Möbius gyromidpoint and Einstein gyromidpoint built upon the framework of gyrovector spaces, respectively, which are represented in different coordinate systems but are geometrically the same, as shown in Appendix D.5. On the other hand, Law et al. (2019) proposed another type of hyperbolic centroid based on the minimization problem of the squared Lorentzian distance defined in the hyperboloid model. Based on the above situation, for the major concern of which formulation to utilize, we proved the following theorem.

**Theorem 1.** *(The equivalence of three hyperbolic midpoints) The Möbius gyromidpoint, Einstein gyromidpoint, and the centroid of the squared Lorentzian distance exactly match each other, which indicates they are the same midpoint operations projected on each manifold.*

The proofs are given in Appendix D.5 and D.6. Furthermore, based on this equivalence, we can characterize the Möbius gyromidpoint as a minimizer of the weighted sum of calibrated squared gyrometrics, which we proved in Appendix D.7.

With Theorem 1, we can now exploit the Möbius gyromidpoint as a unified option to realize hyperbolic attention mechanisms. Moreover, we further generalized the Möbius gyromidpoint by extending the condition of non-negative weights to entire real values by regarding a negative weight as an additive inverse operation: The midpoint $\bar{\boldsymbol{b}} \in \mathbb{B}_c^n$ of Möbius gyrovectors $\{\boldsymbol{b}_i \in \mathbb{B}_c^n\}_{i=1}^N$ with the real scalar weights $\{\nu_i \in \mathbb{R}\}_{i=1}^N$ is given by

$$\bar{\boldsymbol{b}} = \bigoplus_{i=1}^N [\boldsymbol{b}_i, \nu_i]_c := \frac{1}{2} \otimes_c \left( \frac{\sum_{i=1}^N \nu_i \ \lambda_{\boldsymbol{b}_i}^c \boldsymbol{b}_i}{\sum_{i=1}^N |\nu_i| \left( \lambda_{\boldsymbol{b}_i}^c - 1 \right)} \right), \tag{8}$$

which is shown in Appendix D.8. Note that the sum of weights does not need to be normalized to one because any scalar scale is cancelled between the numerator and denominator. On the basis of this insight, in the following, we describe the construction of a multi-head attention as a specific example, aiming at a general approach that can be applied to other arbitrary attention schemes.

**Multi-head attention.** Given a source sequence $\boldsymbol{S} \in \mathbb{R}^{L_s \times n}$ of length $L_s$ and target sequence $\boldsymbol{T} \in \mathbb{R}^{L_t \times m}$ of length $L_t$, the module first projects the target onto query $\boldsymbol{Q} \in \mathbb{R}^{L_t \times hd}$ and the source onto key $\boldsymbol{K} \in \mathbb{R}^{L_s \times hd}$ and value $\boldsymbol{V} \in \mathbb{R}^{L_s \times hd}$ with the corresponding FC layers. These are split into $d$-dimensional vectors of $h$ heads, which is followed by a similarity function between $\boldsymbol{Q}^i$ and $\boldsymbol{K}^i$ producing a weight $\boldsymbol{\Pi}^i = \{\mathrm{softmax}(d^{-\frac{1}{2}} \boldsymbol{q}_t^{i\top} \boldsymbol{K}^i)\}_{t=1}^{L_t}$ for $1 \leq i \leq h$. The weights are utilized to aggregate $\boldsymbol{V}^i$ into a centroid, giving $\boldsymbol{X}^i = \boldsymbol{\Pi}^i \boldsymbol{V}^i$. Finally, the features in all heads are concatenated.

**Poincaré multi-head attention.** Given the source and target as sequences of gyrovectors, they are projected with three Poincaré FC layers, followed by Poincaré $\beta$-splits to produce $\boldsymbol{Q}^i = \{\boldsymbol{q}_t^i \in \mathbb{B}_c^d\}_{t=1}^{L_t}$, $\boldsymbol{K}^i = \{\boldsymbol{k}_s^i \in \mathbb{B}_c^d\}_{s=1}^{L_s}$ and $\boldsymbol{V}^i = \{\boldsymbol{v}_s^i \in \mathbb{B}_c^d\}_{s=1}^{L_s}$. Applying a similarity function $f^c$ and activation $g$, each weight $\pi_{t,s}^i = g(f^c(\boldsymbol{q}_t^i, \boldsymbol{k}_s^i))$ is obtained and the values are aggregated as follows: $\boldsymbol{x}_t^i = \bigoplus_{1 \leq s \leq L_s} [\boldsymbol{v}_s^i, \pi_{t,s}^i]_c$. Finally, the features in all heads are Poincaré $\beta$-concatenated.

For the similarity function $f^c$, there are mainly two choices to exploit. One is the inner product in the tangent space indicated by Micic & Chu (2018), which is the naive generalization of the Euclidean version. Another choice is based on the distance of two points: $f^c(\boldsymbol{q}, \boldsymbol{k}) = -\tau d^c(\boldsymbol{q}, \boldsymbol{k}) - \gamma$, where $\tau$ is an inverse temperature and $\gamma$ is a bias parameter, which was proposed by Gulcehre et al. (2018). As for the activation $g$, $g(x) = \exp(x)$ is the most basic option because it turns to be a softmax operation due to the property of gyromidpoint. Gulcehre et al. (2018) also suggested $g(x) = \sigma(x)$. In light of the property of the generalized gyromidpoint, $g$ as an identity function is also exploitable.

Table 1: Test F1 scores for four sub-trees of the WordNet noun hierarchy. The first column indicates the number of nodes in each sub-tree for the training and test times. For each setting, we report the 95% confidence intervals for three different trials. Note that the number of parameters of the Euclidean MLR and our approach is $D + 1$, whereas for the MLR layer of HNNs, it is $2D$.

| RootNode | Model | D=2 | D=3 | D=5 | D=10 |
|---|---|---|---|---|---|
| animal.n.01 3218 / 798 | Unidirectional (ours) | $\mathbf{60.69}_{\pm \mathbf{4.05}}$ | $67.88_{\pm 1.18}$ | $\mathbf{86.26}_{\pm \mathbf{4.66}}$ | $99.15_{\pm 0.46}$ |
| | HNNs | $59.25_{\pm 16.88}$ | $\mathbf{70.59}_{\pm \mathbf{1.38}}$ | $85.89_{\pm 3.77}$ | $\mathbf{99.34}_{\pm \mathbf{0.39}}$ |
| | Euclidean | $39.96_{\pm 0.89}$ | $60.20_{\pm 0.89}$ | $66.20_{\pm 2.11}$ | $98.33_{\pm 1.12}$ |
| group.n.01 6649 / 1727 | Unidirectional (ours) | $74.27_{\pm 1.50}$ | $63.90_{\pm 6.46}$ | $84.36_{\pm 1.79}$ | $85.60_{\pm 2.75}$ |
| | HNNs | $\mathbf{76.69}_{\pm \mathbf{1.82}}$ | $\mathbf{66.79}_{\pm \mathbf{1.12}}$ | $\mathbf{84.44}_{\pm \mathbf{1.88}}$ | $\mathbf{86.87}_{\pm \mathbf{1.26}}$ |
| | Euclidean | $47.65_{\pm 0.65}$ | $55.15_{\pm 0.97}$ | $71.21_{\pm 1.81}$ | $81.01_{\pm 1.81}$ |
| mammal.n.01 953 / 228 | Unidirectional (ours) | $\mathbf{63.48}_{\pm \mathbf{3.76}}$ | $94.98_{\pm 3.87}$ | $\mathbf{99.30}_{\pm \mathbf{0.30}}$ | $\mathbf{99.17}_{\pm \mathbf{1.55}}$ |
| | HNNs | $46.96_{\pm 13.86}$ | $\mathbf{95.18}_{\pm \mathbf{4.19}}$ | $98.89_{\pm 1.29}$ | $98.75_{\pm 0.51}$ |
| | Euclidean | $15.78_{\pm 0.66}$ | $36.88_{\pm 3.83}$ | $60.53_{\pm 3.27}$ | $65.63_{\pm 2.93}$ |
| location.n.01 2689 / 673 | Unidirectional (ours) | $\mathbf{42.60}_{\pm \mathbf{2.69}}$ | $\mathbf{66.70}_{\pm \mathbf{2.67}}$ | $\mathbf{78.18}_{\pm \mathbf{5.96}}$ | $\mathbf{92.34}_{\pm \mathbf{1.84}}$ |
| | HNNs | $42.57_{\pm 5.03}$ | $62.21_{\pm 26.44}$ | $77.26_{\pm 2.02}$ | $85.14_{\pm 2.86}$ |
| | Euclidean | $34.50_{\pm 0.34}$ | $31.44_{\pm 0.76}$ | $63.86_{\pm 2.18}$ | $82.99_{\pm 3.35}$ |

## 4 EXPERIMENTS

In this section, we evaluate our methods in comparisons with HNNs and Euclidean counterparts. The implementation of hyperbolic architectures is based on the Geoopt (Kochurov et al., 2020).

### 4.1 VERIFICATION OF THE MLR CLASSIFICATION CAPACITY

We first evaluated the performance of our unidirectional Poincaré MLR on the same conditioned experiment designed for the MLR of HNNs, that is, a sub-tree classification on the Poincaré ball model. In this task, the Poincaré embeddings of the WordNet noun hierarchy (Nickel & Kiela, 2017) are utilized as the data set, which contains 82,115 nodes and 743,241 hypernymy relations. We pre-trained the Poincaré embeddings of the same dimensions as the experimental settings in HNNs, *i.e.*, two, three, five, and ten dimensions, using the open-source implementation[2] to extract several sub-trees whose root nodes are certain abstract hypernymies, *e.g.*, animal. For each sub-tree, MLR layers learn the binary classification to predict whether each given node is included. All nodes are divided into 80% training nodes and 20% testing nodes. We trained each model for 30 epochs using Riemannian Adam (Becigneul & Ganea, 2019) with a learning rate of 0.001 and a batch size of 16.

The F1 scores for the test sets are shown in Table 1. From the results, we can confirm the tendency of the hyperbolic MLRs to outperform the Euclidean version in all settings, which illustrates that MLR considering the hyperbolic geometry are better suited to the hyperbolic embeddings. In particular, our parameter-reduced approach obtains the same level of performance as a conventional hyperbolic MLR in a more stable training, as can be seen from the relatively narrower confidence intervals.

### 4.2 AMORTIZED CLUSTERING OF MIXTURE OF GAUSSIANS WITH SET TRANSFORMERS

For the evaluation of a Poincaré multi-head attention, we utilize Set Transformer, which we consider is a proper test case to eliminate the implicit influence of unessential operations, *e.g.*, positional encoding. The task is an amortized clustering of a mixture of Gaussians (MoG). In each sample in a mini-batch, models take hundreds of two-dimensional points randomly generated by the same $K$-component MoG, and directly estimate all the parameters, *i.e.*, the ground truth probabilities, means, and standard deviations, in a single forward step. We basically follow the model architectures and experimental settings of the official implementation[3], except that we employed the hyperbolic Gaussian distribution (Ovinnikov, 2019) as well as the Euclidean distribution aiming to verify the performance of the hyperbolic architectures both for the ordinary settings and for their desirable data

---

[2] https://github.com/facebookresearch/poincare-embeddings
[3] https://github.com/juho-lee/set_transformer

distributions. When the hyperbolic models need to deal with Euclidean coordinates, the inputs or outputs are projected by an exponential map or logarithmic map, respectively, with a scalar parameter for scaling the values to the fixed-radius Poincaré ball $\mathbb{B}_1^n$. Note that we omit ReLU activations for our models because the hyperbolic operations are inherently non-linear. We also remove normalization layers because there does not exist enough research on the normalizing criterion in the hyperbolic space and existing methods possibly reduce the representational power of gyrovectors by neutralizing their norms. For the hyperbolic attentions, based on a preliminary experiment, we choose to utilize the distance based similarity function and exponential activation.

Table 2: Negative log-likelihood on the test set. For each setting, we report the 95% confidence intervals for five trials. The numbers in brackets indicate the diverged trials, the final scores of which were higher than 10.0, and those trials are not accounted into the reported scores.

| Model | K=4 | K=5 | K=6 | K=7 | K=8 |
|---|---|---|---|---|---|
| Gaussian distribution on Euclidean space | | | | | |
| Set Transformer w/o LN | $\textbf{1.556}_{\pm \textbf{0.214 (3)}}$ | $1.912_{\pm 0.701 (2)}$ | $\textbf{2.032}_{\pm \textbf{0.193 (3)}}$ | $5.066_{\pm 5.239 (3)}$ | $2.608_{\pm \text{N/A (4)}}$ |
| Set Transformer | $1.558_{\pm 0.032 (0)}$ | $\textbf{1.776}_{\pm \textbf{0.030 (0)}}$ | $2.046_{\pm 0.030 (0)}$ | $\textbf{2.297}_{\pm \textbf{0.047 (0)}}$ | $\textbf{2.519}_{\pm \textbf{0.020 (0)}}$ |
| Ours | $1.558_{\pm 0.008 (0)}$ | $1.833_{\pm 0.046 (0)}$ | $2.081_{\pm 0.036 (0)}$ | $2.370_{\pm 0.098 (0)}$ | $2.682_{\pm 0.164 (0)}$ |
| Generalized Gaussian distribution on the Poincaré ball model (Ovinnikov, 2019) | | | | | |
| Set Transformer | $3.084_{\pm 0.305 (0)}$ | $3.298_{\pm 0.414 (0)}$ | $3.327_{\pm 0.316 (0)}$ | $3.923_{\pm 1.632 (0)}$ | $3.519_{\pm 0.125 (0)}$ |
| Ours | $\textbf{2.920}_{\pm \textbf{0.029 (0)}}$ | $\textbf{3.087}_{\pm \textbf{0.014 (0)}}$ | $\textbf{3.252}_{\pm \textbf{0.037 (0)}}$ | $\textbf{3.375}_{\pm \textbf{0.033 (0)}}$ | $\textbf{3.462}_{\pm \textbf{0.013 (0)}}$ |

The results are shown in Table 2. For the Euclidean distribution, our models achieved almost the same performance as Set Transformers, while the training of those without Layer Normalization for the same conditioned comparison often failed under all settings. This result suggests the intrinsic normalization properties of our methods, which we attribute to the computation using vector norms. For the hyperbolic distribution, our models outperformed the Euclidean counterparts with an order of magnitude smaller confidence intervals, which indicates that our hyperbolic architectures are indeed suited to their assumed data distribution.

## 4.3 CONVOLUTIONAL SEQUENCE TO SEQUENCE MODELING

Finally, we experimented with the convolutional sequence-to-sequence modeling task for machine translation of WMT'17 English-German (Bojar et al., 2017). Because the architecture is composed of convolutional layers and attention layers, the hyperbolic version of which has already been verified in Section 4.2, it would provide a comparison focusing on convolutional operations. It also has a practical aspect as a task of natural language processing, in which lexical entities are known to form latent hierarchical structures. We follow the open-source implementation of Fairseq (Ott et al., 2019), where preprocessed training data contains 3.96M sentence pairs with 40K sub-word tokenization in each language. In our hyperbolic models, feature vectors are completely treated as Möbius gyrovectors because token embeddings can be learned directly on the Poincaré ball model. Note that the inputs for the sigmoid functions in Gated Linear Units are logarithmically mapped just like hyperbolic Gated Recurrent Units proposed by Ganea et al. (2018a). We train various scaled-down models to verify the representational capacity of our hyperbolic architectures, with Riemannian Adam for 100K iterations. For more implementation details, please check Appendix E.

Table 3: BLEU-4 scores (Papineni et al., 2002) on the test sets newstest2013. The target sentences were decoded using beam search with a beam size of five. $D$ indicates the dimensions of token embeddings and the final MLR layer.

| Model | D=16 | D=32 | D=64 | D=128 | D=256 |
|---|---|---|---|---|---|
| ConvSeq2Seq | 2.68 | 8.43 | 14.92 | **20.02** | **21.84** |
| Ours | **9.81** | **14.11** | **16.95** | 19.40 | 21.76 |

The results are shown in Table 3. Our model demonstrates the significant improvements compared to the usual Euclidean models in the fewer dimensions, which reflects the immense embedding

capacity of hyperbolic spaces. On the other hand, there is no salient differences observed in higher dimensions, which implies that the Euclidean models with higher dimensions than a certain level can obtain a sufficient computational complexity through the optimization. This would fill the gap with the representational properties of hyperbolic spaces. It also implies that the proper construction of neural networks with the product space of multiple small hyperbolic spaces using our methods has the potential for the further improvements even in higher dimensional architectures.

## 5 CONCLUSION

We showed a novel generalization and construction scheme of the wide range of hyperbolic neural network architectures in the Poincaré ball model, including a parameter-reduced MLR, geodesic-aware FC layers, convolutional layers, and attention mechanisms. These were achieved under a unified mathematical backbone based on the concepts of Riemannian geometry and the Möbius gyrovector space, which endow our hyperbolic architectures with theoretical consistency. Through the experiments, we verified the effectiveness of our approaches from diversified tasks and perspectives, such as an embedded sub-tree classification, amortized clustering of distributed points both on the Euclidean space and the Poincaré ball model, and neural machine translation. We hope that this study will pave the way for future research in the field of geometric deep learning.

ACKNOWLEDGMENTS

We would like to thank Naoyuki Gunji and Yuki Kawana from the University of Tokyo for their helpful discussions and constructive advice. We would also like to show our gratitude to Dr. Lin Gu from RIKEN AIP, Kenzo Lobos-Tsunekawa from the University of Tokyo, and Editage[4] for proofreading the manuscript for English language.

This work was partially supported by JST AIP Acceleration Research Grant Number JPMJCR20U3, JST CREST Grant Number JPMJCR2015, JSPS KAKENHI Grant Number JP19H01115, and Basic Research Grant (Super AI) of Institute for AI and Beyond of the University of Tokyo.

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

## A  HYPERBOLIC GEOMETRY

In this section, we review the definition of the hyperbolic geometry models other than the Poincaré ball model and the relationships between their coordinates.

**Hyperboloid model.** The $n$-dimensional hyperboloid model is a hypersurface in an $(n + 1)$-dimensional Minkowski space $\mathbb{R}_1^{n+1}$, which is equipped with an inner product $\langle \boldsymbol{x}, \boldsymbol{y} \rangle_{\mathcal{L}} = \boldsymbol{x}^\top \mathfrak{g}_{\mathcal{L}} \boldsymbol{y}$ for $^\forall \boldsymbol{x}, \boldsymbol{y} \in \mathbb{R}_1^{n+1}$, where $\mathfrak{g}_{\mathcal{L}} = \mathrm{diag}(-1, \mathbf{1}_n^\top)$. Given a constant negative curvature $-c$, the manifold of the hyperboloid model is defined by $\mathbb{H}_c^n = \{\boldsymbol{x} = (x_0, \dots, x_n)^\top \in \mathbb{R}_1^{n+1} \mid c \langle \boldsymbol{x}, \boldsymbol{x} \rangle_{\mathcal{L}} = -1, x_0 > 0\}$.

Note that, in this standard $(n + 1)$-dimensional coordinate system, the metric tensor as a positive definite matrix for the $n$-dimensional hyperboloid manifold cannot be defined. Instead, when the hyperboloid model is represented in a specific $n$-dimensional coordinates, *e.g.*, hyperbolic polar coordinates, then its metric tensor has the corresponding representation as the $n$-dimensional positive definite matrix.

**Isometric isomorphism with the Poincaré ball model.** The bijection between an arbitrary point $\boldsymbol{h} = (z, \boldsymbol{k}^\top)^\top \in \mathbb{H}_c^n$ and its unique corresponding point $\boldsymbol{b} \in \mathbb{B}_c^n$, depicted in Figure 1, is given by the following:

$$\mathbb{H}_c^n \to \mathbb{B}_c^n : \ \boldsymbol{b} = \boldsymbol{b}(\boldsymbol{h}) = \frac{\boldsymbol{k}}{1 + \sqrt{c}z}, \tag{9}$$

$$\mathbb{B}_c^n \to \mathbb{H}_c^n : \ \boldsymbol{h} = \boldsymbol{h}(\boldsymbol{b}) = (z(\boldsymbol{b}), \boldsymbol{k}(\boldsymbol{b})) = \left( \frac{1}{\sqrt{c}} \frac{1 + c\|\boldsymbol{b}\|^2}{1 - c\|\boldsymbol{b}\|^2}, \frac{2\boldsymbol{b}}{1 - c\|\boldsymbol{b}\|^2} \right). \tag{10}$$

**Beltrami-Klein model.** The $n$-dimensional Beltrami-Klein model of a constant negative curvature $-c$ is defined by $(\mathbb{K}_c^n, \hat{\mathfrak{g}}^c)$, where $\mathbb{K}_c^n = \{\boldsymbol{x} \in \mathbb{R}^n \mid c\|\boldsymbol{x}\|^2 < 1\}$ and $\hat{\mathfrak{g}}_{\boldsymbol{x}}^c = (1 - c\|\boldsymbol{x}\|^2)^{-1} \boldsymbol{I}_n + (1 - c\|\boldsymbol{x}\|^2)^{-2} \boldsymbol{x}\boldsymbol{x}^\top$. Here, $\mathbb{K}_c^n$ is an open ball of radius $1/\sqrt{c}$.

**Isometric isomorphism with the Poincaré ball model.** The bijection between an arbitrary point $\boldsymbol{n} \in \mathbb{K}_c^n$ and its unique corresponding point $\boldsymbol{b} \in \mathbb{B}_c^n$, depicted in Figure 1, is given by the following:

$$\mathbb{K}_c^n \to \mathbb{B}_c^n : \ \boldsymbol{b} = \boldsymbol{b}(\boldsymbol{n}) = \frac{\boldsymbol{n}}{1 + \sqrt{1 - c\|\boldsymbol{n}\|^2}}, \tag{11}$$

$$\mathbb{B}_c^n \to \mathbb{K}_c^n : \ \boldsymbol{n} = \boldsymbol{n}(\boldsymbol{b}) = \frac{2\boldsymbol{b}}{1 + c\|\boldsymbol{b}\|^2}. \tag{12}$$

## B  MÖBIUS GYROVECTOR SPACE

In this section, we briefly introduce the concept of the Möbius gyrovector space, which is a specific type of gyrovector spaces. For a rigorous theoretical and detailed mathematical background of this system, please refer to Ungar (2005; 2009; 2001; 2012).

A gyrovector space is an algebraic structure that endows the points in a hyperbolic space with vector-like properties based on a special concept called a gyrogroup. This gyrogroup is similar to ordinary vector spaces that provides a Euclidean space with the well-known vector operations based on the notion of groups. As a particular example in physics, this helps to understand the mathematical structure of the Einstein's theory of special relativity where no possible velocity vectors including the sum of velocities in an arbitrary additive order can exceed the speed of light (Ungar, 2008; 2013). Because hyperbolic geometry has several isometric models, a gyrovector space also has some variants where the Möbius gyrovector space is a variant for the Poincaré ball model.

As an abstract mathematical system, a gyrovector space is constructed through the following steps: (1) Start from a set $G$. (2) With a certain binary operation $\oplus$, create a tuple called a groupoid, or magma $(G, \oplus)$. (3) Based on five axioms, define a specific type of magma as a gyrogroup. These axioms include several important properties of gyrovector spaces, such as the left gyroassociative law and an operator called a gyrator $\mathrm{gyr} : G \times G \to \mathrm{Aut}(G, \oplus)$, which generates an automorphism $\mathrm{Aut}(G, \oplus) \ni \mathrm{gyr}[\boldsymbol{x}, \boldsymbol{y}] : G \to G$ given by $\boldsymbol{z} \mapsto \mathrm{gyr}[\boldsymbol{x}, \boldsymbol{y}]\boldsymbol{z}$, called a gyration, from two arbitrary points $\boldsymbol{x}$ and $\boldsymbol{y} \in G$. The notion of the gyrocommutative law and gyrogroup cooperation are given in this step. (4) Adding ten more axioms related to the statements about a real inner product space and a scalar multiplication $\otimes$, the gyrovector space $(G, \oplus, \otimes)$ is thus defined.

Some of the important properties of a gyrovector space are listed below. Here, $\boldsymbol{x}, \boldsymbol{y}, \boldsymbol{z} \in G$.

**Gyroassociative laws.** Although the binary operation $\oplus$ is not necessarily associative in general, it obeys the left gyroassociative law $\boldsymbol{x} \oplus (\boldsymbol{y} \oplus \boldsymbol{z}) = (\boldsymbol{x} \oplus \boldsymbol{y}) \oplus \mathrm{gyr}[\boldsymbol{x}, \boldsymbol{y}]\boldsymbol{z}$ and right gyroassociative law $(\boldsymbol{x} \oplus \boldsymbol{y}) \oplus \boldsymbol{z} = \boldsymbol{x} \oplus (\boldsymbol{y} \oplus \mathrm{gyr}[\boldsymbol{y}, \boldsymbol{x}]\boldsymbol{z})$. These equations also provide a general closed-form expression of the gyrations: $\mathrm{gyr}[\boldsymbol{x}, \boldsymbol{y}]\boldsymbol{z} = \ominus(\boldsymbol{x} \oplus \boldsymbol{y}) \oplus (\boldsymbol{x} \oplus (\boldsymbol{y} \oplus \boldsymbol{z}))$.

**Cases in which gyrations become identity maps.** If at least one element for gyr is $\boldsymbol{0} \in G$, the gyrations become an identity map $I$: $\mathrm{gyr}[\boldsymbol{x}, \boldsymbol{0}] = \mathrm{gyr}[\boldsymbol{0}, \boldsymbol{x}] = I$. With the loop properties of the gyrations given by $\mathrm{gyr}[\boldsymbol{x}, \boldsymbol{y}] = \mathrm{gyr}[\boldsymbol{x} \oplus \boldsymbol{y}, \boldsymbol{y}] = \mathrm{gyr}[\boldsymbol{x}, \boldsymbol{y} \oplus \boldsymbol{x}]$, many other cases can be also derived.

**Gyrocommutative law.** Although a binary operation $\oplus$ is not necessarily commutative in general, if it obeys the equation $\boldsymbol{x} \oplus \boldsymbol{y} = \mathrm{gyr}[\boldsymbol{x}, \boldsymbol{y}](\boldsymbol{y} \oplus \boldsymbol{x})$, the gyrogroup is called gyrocommutative.

**Gyrogroup cooperation.** Regarding $\oplus$ as the primal binary addition, the second binary addition in $G$ is defined as the gyrogroup cooperation $\boxplus$, which is given by $\boldsymbol{x} \boxplus \boldsymbol{y} = \boldsymbol{x} \oplus \mathrm{gyr}[\boldsymbol{x}, \ominus\boldsymbol{y}]\boldsymbol{y}$. This has duality symmetries with the first binary operation $\oplus$, such that $\boldsymbol{x} \oplus \boldsymbol{y} = \boldsymbol{x} \boxplus \mathrm{gyr}[\boldsymbol{x}, \boldsymbol{y}]\boldsymbol{y}$. In addition, corresponding to the left cancellation law $\ominus\boldsymbol{x} \oplus (\boldsymbol{x} \oplus \boldsymbol{y}) = \boldsymbol{y}$ inherent in $\oplus$, the gyrogroup cooperation induces two types of the right cancellation laws: $(\boldsymbol{x} \oplus \boldsymbol{y}) \boxminus \boldsymbol{y} = (\boldsymbol{x} \boxplus \boldsymbol{y}) \ominus \boldsymbol{y} = \boldsymbol{x}$.

In this formalism, the Möbius gyrovector space is then defined as $(\mathbb{B}_c^n, \oplus_c, \otimes_c)$, where $\mathbb{B}_c^n$ is as previously introduced in Section 2, and $\oplus_c$ and $\otimes_c$ are as shown in the following subsections.

## B.1 Möbius addition

In the Möbius gyrovector space, the primary binary operation is denoted as the Möbius addition $\oplus_c : \mathbb{B}_c^n \times \mathbb{B}_c^n \to \mathbb{B}_c^n$, which is a noncommutaive and nonassociative addition, given by the following:

$$\boldsymbol{x} \oplus_c \boldsymbol{y} = \frac{\left(1 + 2c\langle \boldsymbol{x}, \boldsymbol{y}\rangle + c\|\boldsymbol{y}\|^2\right)\boldsymbol{x} + \left(1 - c\|\boldsymbol{x}\|^2\right)\boldsymbol{y}}{1 + 2c\langle \boldsymbol{x}, \boldsymbol{y}\rangle + c^2\|\boldsymbol{x}\|^2\|\boldsymbol{y}\|^2}, \quad \boldsymbol{x} \ominus_c \boldsymbol{y} = \boldsymbol{x} \oplus_c (-\boldsymbol{y}). \tag{13}$$

## B.2 Möbius gyrator

The expression of gyrations in the Möbius gyrovector space can be expanded using the equation of the Möbius addition $\oplus_c$, which is described by Ungar (2009) as follows:

$$\mathrm{gyr}[\boldsymbol{x}, \boldsymbol{y}] : \boldsymbol{z} \mapsto \boldsymbol{z} - 2c\frac{\left(c\langle \boldsymbol{x}, \boldsymbol{z}\rangle\|\boldsymbol{y}\|^2 - \langle \boldsymbol{y}, \boldsymbol{z}\rangle\left(1 + 2c\langle \boldsymbol{x}, \boldsymbol{y}\rangle\right)\right)\boldsymbol{x} + \left(c\langle \boldsymbol{y}, \boldsymbol{z}\rangle\|\boldsymbol{x}\|^2 + \langle \boldsymbol{x}, \boldsymbol{z}\rangle\right)\boldsymbol{y}}{1 + 2c\langle \boldsymbol{x}, \boldsymbol{y}\rangle + c^2\|\boldsymbol{x}\|^2\|\boldsymbol{y}\|^2}.$$
$$\tag{14}$$

By writing down all the special operators $\oplus_c$ for the gyrovectors in $\mathbb{B}_c^n$ into the normal vector operations, the expression of the gyrations can be now seen as a general function for the any real vector $\boldsymbol{z} \in \mathbb{R}^n$. Indeed, gyrations are extended to invertible linear maps of $\mathbb{R}^n$ (Ungar, 2009).

The Möbius gyrator endows the Möbius gyrovector space with a gyrocommutative nature.

## B.3 Möbius coaddition

The gyrogroup cooperation in the Möbius gyrovector space is called the Möbius coaddition, and is given by the following:

$$\boldsymbol{x} \boxplus_c \boldsymbol{y} = \boldsymbol{x} \oplus_c \mathrm{gyr}[\boldsymbol{x}, \ominus_c\boldsymbol{y}]\boldsymbol{y} = \frac{\left(1 - c\|\boldsymbol{y}\|^2\right)\boldsymbol{x} + \left(1 - c\|\boldsymbol{x}\|^2\right)\boldsymbol{y}}{1 - c^2\|\boldsymbol{x}\|^2\|\boldsymbol{y}\|^2}.$$

With the gamma factor $\gamma_{\boldsymbol{x}} = (\sqrt{1 - c\|\boldsymbol{x}\|^2})^{-1}$ for $\boldsymbol{x} \in \mathbb{B}_c^n$, this is also described in the following manner:

$$\boldsymbol{x} \boxplus_c \boldsymbol{y} = \frac{\gamma_{\boldsymbol{x}}^2\boldsymbol{x} + \gamma_{\boldsymbol{y}}^2\boldsymbol{y}}{\gamma_{\boldsymbol{x}}^2 + \gamma_{\boldsymbol{y}}^2 - 1}. \tag{15}$$

Note that the Möbius coaddition is not associative but is commutative.

### B.4 MÖBIUS SCALAR MULTIPLICATION

The Möbius scalar multiplication for $\boldsymbol{x} \in \mathbb{B}_c^n$ and $r \in \mathbb{R}$ is given by the following:

$$r \otimes_c \boldsymbol{x} = \frac{1}{\sqrt{c}} \tanh^{-1}\left(r \tanh\left(\sqrt{c}\,\|\boldsymbol{x}\|\right)\right)[\boldsymbol{x}] = \exp_{\boldsymbol{0}}^c\left(r \log_{\boldsymbol{0}}^c(\boldsymbol{x})\right). \tag{16}$$

In terms of the Riemannian geometry, the Möbius scalar multiplication adjusts the distance of $\boldsymbol{x}$ from the origin by the scalar multiplier $r$. The expressions of the logarithmic map $\log_{\boldsymbol{x}}^c$ and distance in the Möbius gyrovector space are described in the following subsections.

## C   POINCARÉ BALL MODEL

Owing to the algebraic structure provided by the Möbius gyrovector space, many properties related to the geometry of the Poincaré ball model can be described in implementation-friendly closed-form expressions.

### C.1   EXPONENTIAL AND LOGARITHMIC MAPS

The exponential map $\exp_{\boldsymbol{x}}^c : \mathcal{T}_{\boldsymbol{x}}\mathbb{B}_c^n \to \mathbb{B}_c^n$ is described in (Ganea et al., 2018a, Lemma 2) as follows:

$$\exp_{\boldsymbol{x}}^c(\boldsymbol{v}) = \boldsymbol{x} \oplus_c \frac{1}{\sqrt{c}} \tanh\left(\frac{\sqrt{c}\lambda_{\boldsymbol{x}}^c\|\boldsymbol{v}\|}{2}\right)[\boldsymbol{v}], \quad \forall \boldsymbol{x} \in \mathbb{B}_c^n,\ \boldsymbol{v} \in \mathcal{T}_{\boldsymbol{x}}\mathbb{B}_c^n. \tag{17}$$

The logarithmic map $\log_{\boldsymbol{x}}^c = (\exp_{\boldsymbol{x}}^c)^{-1} : \mathbb{B}_c^n \to \mathcal{T}_{\boldsymbol{x}}\mathbb{B}_c^n$ is also given by the following:

$$\log_{\boldsymbol{x}}^c(\boldsymbol{y}) = \frac{2}{\sqrt{c}\lambda_{\boldsymbol{x}}^c} \tanh^{-1}\left(\sqrt{c}\|\ominus_c \boldsymbol{x} \oplus_c \boldsymbol{y}\|\right)[\ominus_c\boldsymbol{x} \oplus_c \boldsymbol{y}], \quad \forall \boldsymbol{x},\ \boldsymbol{y} \in \mathbb{B}_c^n. \tag{18}$$

### C.2   DISTANCE

#### C.2.1   POINCARÉ DISTANCE BETWEEN TWO ARBITRARY POINTS

The distance function $d_c$ is originally and preliminary defined as a binary operation for indicating the distance between two arbitrary points $\boldsymbol{x}, \boldsymbol{y} \in \mathbb{B}_c^n$. Based on the notion of the Möbius addition, the distance $d_c : \mathbb{B}_c^n \times \mathbb{B}_c^n \to \mathbb{R}$ is succinctly described as follows:

$$d_c(\boldsymbol{x}, \boldsymbol{y}) = \frac{2}{\sqrt{c}} \tanh^{-1}\left(\sqrt{c}\|\ominus_c \boldsymbol{x} \oplus_c \boldsymbol{y}\|\right) = \|\log_{\boldsymbol{x}}^c(\boldsymbol{y})\|_{\boldsymbol{x}}^c. \tag{19}$$

Despite the noncommutative aspect of the Möbius addition $\oplus_c$, this distance function in Equation 19 becomes commutative thanks to the commutative aspect of the Euclidean norm of the Möbius addition, which is expressed as follows:

$$\|\boldsymbol{x} \oplus_c \boldsymbol{y}\| = \sqrt{\frac{\|\boldsymbol{x}\|^2 + 2\langle\boldsymbol{x}, \boldsymbol{y}\rangle + \|\boldsymbol{y}\|^2}{1 + 2c\langle\boldsymbol{x}, \boldsymbol{y}\rangle + c^2\|\boldsymbol{x}\|^2\|\boldsymbol{y}\|^2}}, \quad \forall\ \boldsymbol{x}, \boldsymbol{y} \in \mathbb{B}_c^n. \tag{20}$$

#### C.2.2   DISTANCE FROM A POINT TO POINCARÉ HYPERPLANE

In the Euclidean geometry, the generalized concept of two-dimensional plane to a higher dimensional space $\mathbb{R}^n$ is a hyperplane containing an arbitrary point $\boldsymbol{p} \in \mathbb{R}^n$ and is the set of all straight lines orthogonal to an arbitrary orientation vector $\boldsymbol{a} \in \mathbb{R}^n$. Because straight lines in Euclidean spaces are geodesics in terms of the Riemannian geometry, a hyperplane can be generalized as another Riemannian manifold $\mathcal{M}^n$ such that the hyperplane contains an arbitrary point $\boldsymbol{p} \in \mathcal{M}^n$ and is the set of all geodesics orthogonal to an arbitrary orientation vector at $\boldsymbol{p}$, namely, the tangent vector $\boldsymbol{a} \in \mathcal{T}_{\boldsymbol{p}}\mathcal{M}^n$. This concept in the Poincaré ball model has been rigorously defined in (Ganea et al., 2018a, Definition 3.1) for $\boldsymbol{p} \in \mathbb{B}_c^n, \boldsymbol{a} \in \mathcal{T}_{\boldsymbol{p}}\mathbb{B}_c^n$ as follows:

$$\tilde{H}_{\boldsymbol{a},\boldsymbol{p}}^c = \{\boldsymbol{x} \in \mathbb{B}_c^n \mid \langle\log_{\boldsymbol{p}}^c(\boldsymbol{x}), \boldsymbol{a}\rangle_{\boldsymbol{p}}^c = 0\} = \exp_{\boldsymbol{p}}^c\left(\{\boldsymbol{a}\}^\perp\right) \tag{21}$$

$$= \{\boldsymbol{x} \in \mathbb{B}_c^n \mid \langle\ominus_c\boldsymbol{p} \oplus_c \boldsymbol{x}, \boldsymbol{a}\rangle = 0\}. \tag{22}$$

Note that $\{\boldsymbol{a}\}^{\perp}$ is the set of all tangent vectors at $\boldsymbol{p}$ and orthogonal to $\boldsymbol{a}$.

Ganea et al. (2018a) have proven the closed-form description of the distance from a point $\boldsymbol{x} \in \mathbb{B}_c^n$ to an arbitrary Poincaré hyperplane $\tilde{H}_{\boldsymbol{a},\boldsymbol{p}}^c$ by considering the minimum distance between $\boldsymbol{x}$ and any point in $\tilde{H}_{\boldsymbol{a},\boldsymbol{p}}^c$:

$$d_c\left(\boldsymbol{x}, \tilde{H}_{\boldsymbol{a},\boldsymbol{p}}^c\right) \coloneqq \inf_{\boldsymbol{w} \in \tilde{H}_{\boldsymbol{a},\boldsymbol{p}}^c} d_c\left(\boldsymbol{x}, \boldsymbol{w}\right) = \frac{1}{\sqrt{c}} \sinh^{-1}\left(\frac{2\sqrt{c}|\langle \ominus_c \boldsymbol{p} \oplus_c \boldsymbol{x}, \boldsymbol{a}\rangle|}{(1 - c\|\ominus_c \boldsymbol{p} \oplus_c \boldsymbol{x}\|^2)\|\boldsymbol{a}\|}\right). \tag{23}$$

### C.3 Parallel transport

The concept of a parallel transport is traditionally derived from differential geometry. In the hyperbolic geometry, the gyrovector space provides the algebra to formulate the parallel transport of a gyrovector (Ungar, 2012). When a gyrovector $\ominus_c \boldsymbol{x} \oplus_c \boldsymbol{w} \in \mathbb{B}_c^n$ rooted at a point $\boldsymbol{x} \in \mathbb{B}_c^n$ is transported parallel to another gyrovector $\ominus_c \boldsymbol{y} \oplus_c \boldsymbol{z} \in \mathbb{B}_c^n$ rooted at a point $\boldsymbol{y} \in \mathbb{B}_c^n$ along a geodesic connecting $\boldsymbol{x}$ and $\boldsymbol{y}$, the equation below is satisfied:

$$\ominus_c \boldsymbol{y} \oplus_c \boldsymbol{z} = \mathrm{gyr}[\boldsymbol{y}, \ominus_c \boldsymbol{x}] \left(\ominus_c \boldsymbol{x} \oplus_c \boldsymbol{w}\right). \tag{24}$$

Because the exponential map in the Poincaré ball model is a bijective function, the parallel transported gyrovectors $\boldsymbol{w}$ and $\boldsymbol{z}$ can be regarded as the exponential mapped tangent vectors $\boldsymbol{v} \in \mathcal{T}_{\boldsymbol{x}} \mathbb{B}_c^n$ rooted at $\boldsymbol{x}$ and $\boldsymbol{u} \in \mathcal{T}_{\boldsymbol{y}} \mathbb{B}_c^n$ rooted at $\boldsymbol{y}$, respectively, that is,

$$\ominus_c \boldsymbol{y} \oplus_c \exp_{\boldsymbol{y}}^c\left(\boldsymbol{u}\right) = \mathrm{gyr}[\boldsymbol{y}, \ominus_c \boldsymbol{x}]\left(\ominus_c \boldsymbol{x} \oplus_c \exp_{\boldsymbol{x}}^c\left(\boldsymbol{v}\right)\right). \tag{25}$$

With Equations 17 and 18 and the properties of the Möbius gyration described in Appendix B, a succinct expression of the tangent parallel transport $P_{\boldsymbol{x} \to \boldsymbol{y}}^c : \mathcal{T}_{\boldsymbol{x}} \mathbb{B}_c^n \to \mathcal{T}_{\boldsymbol{y}} \mathbb{B}_c^n$ can be obtained as follows:

$$P_{\boldsymbol{x} \to \boldsymbol{y}}^c(\boldsymbol{v}) \coloneqq \boldsymbol{u} = \frac{\lambda_{\boldsymbol{x}}^c}{\lambda_{\boldsymbol{y}}^c} \mathrm{gyr}[\boldsymbol{y}, \ominus_c \boldsymbol{x}] \boldsymbol{v}. \tag{26}$$

Note that, in a special case in which $\boldsymbol{x} = \boldsymbol{0}$ and $\boldsymbol{v} \in \mathcal{T}_{\boldsymbol{0}} \mathbb{B}_c^n$, this equation is simplified as follows:

$$P_{\boldsymbol{0} \to \boldsymbol{y}}^c(\boldsymbol{v}) = \frac{\lambda_{\boldsymbol{0}}^c}{\lambda_{\boldsymbol{y}}^c} \boldsymbol{v} = \left(1 - c\|\boldsymbol{y}\|^2\right) \boldsymbol{v}. \tag{27}$$

One can confirm that Equation 26 deserves to be called a parallel transport in terms of the differential or Riemannian geometry by checking the covariant derivative associated with the Levi-Civita connection of $P_{\boldsymbol{x} \to \boldsymbol{y}}^c$ along a tangent vector field $\dot{\gamma}(t)$ on a smooth curve $\gamma(t)$ from $\boldsymbol{x}$ to $\boldsymbol{y}$ vanishes to $\boldsymbol{0}$.

## D Supplemental proofs for proposed methods

### D.1 Final deformation of the proposed unidirectional Poincaré MLR

*Proof.* First, we clarify the relation between the Poincaré hyperplane $\tilde{H}_{\boldsymbol{a},\boldsymbol{p}}^c$, described in Appendix C.2.2, and the variants $\bar{H}_{\boldsymbol{a},r}^c$ introduced in Section 3.1:

$$\bar{H}_{\boldsymbol{a},r}^c = \tilde{H}_{\boldsymbol{a}, \boldsymbol{q}_{\boldsymbol{a}_k, r_k}}^c. \tag{28}$$

We then start the derivation of Equation 6 from the variables $\boldsymbol{a}_k$ and $\boldsymbol{q}_{\boldsymbol{a}_k, r_k}$ described in Section 3.1. Following Equation 28 and the concept of the distance from a point to a Poincaré hyperplane described in Equation 23, the generalized MLR score function $v_k$ in Equation 3 can be written as follows:

$$v_k(\boldsymbol{x}) = \frac{\lambda_{\boldsymbol{q}_{\boldsymbol{a}_k, r_k}}^c \|\boldsymbol{a}_k\|}{\sqrt{c}} \sinh^{-1}\left(\frac{2\sqrt{c}\langle \ominus_c \boldsymbol{q}_{\boldsymbol{a}_k, r_k} \oplus_c \boldsymbol{x}, \boldsymbol{a}_k \rangle}{(1 - c\|\ominus_c \boldsymbol{q}_{\boldsymbol{a}_k, r_k} \oplus_c \boldsymbol{x}\|^2)\|\boldsymbol{a}_k\|}\right), \quad \forall \boldsymbol{x} \in \mathbb{B}_c^n. \tag{29}$$

With Equation 20, we obtain

$$\|\ominus_c \boldsymbol{q}_{\boldsymbol{a}_k, r_k} \oplus_c \boldsymbol{x}\|^2 = \frac{\|\boldsymbol{x}\|^2 - 2\langle \boldsymbol{x}, \boldsymbol{q}_{\boldsymbol{a}_k, r_k}\rangle + \|\boldsymbol{q}_{\boldsymbol{a}_k, r_k}\|^2}{1 - 2c\langle \boldsymbol{x}, \boldsymbol{q}_{\boldsymbol{a}_k, r_k}\rangle + c^2\|\boldsymbol{x}\|^2\|\boldsymbol{q}_{\boldsymbol{a}_k, r_k}\|^2}. \tag{30}$$

Therefore, we can expand the term inside the $\sinh^{-1}$ function in Equation 29 in the following manner:

$$
\frac{2\sqrt{c}\langle \ominus_c \boldsymbol{q}_{\boldsymbol{a}_k,r_k} \oplus_c \boldsymbol{x}, \boldsymbol{a}_k\rangle}{(1 - c\|\ominus_c \boldsymbol{q}_{\boldsymbol{a}_k,r_k} \oplus_c \boldsymbol{x}\|^2)\|\boldsymbol{a}_k\|}
$$

$$
= \frac{2\sqrt{c}}{\|\boldsymbol{a}_k\|}\frac{-\left(1 - 2c\langle \boldsymbol{x}, \boldsymbol{q}_{\boldsymbol{a}_k,r_k}\rangle + c\|\boldsymbol{x}\|^2\right)\langle \boldsymbol{q}_{\boldsymbol{a}_k,r_k}, \boldsymbol{a}_k\rangle + \left(1 - c\|\boldsymbol{q}_{\boldsymbol{a}_k,r_k}\|^2\right)\langle \boldsymbol{x}, \boldsymbol{a}_k\rangle}{1 - 2c\langle \boldsymbol{x}, \boldsymbol{q}_{\boldsymbol{a}_k,r_k}\rangle + c^2\|\boldsymbol{x}\|^2\|\boldsymbol{q}_{\boldsymbol{a}_k,r_k}\|^2 - c\left(\|\boldsymbol{x}\|^2 - 2\langle \boldsymbol{x}, \boldsymbol{q}_{\boldsymbol{a}_k,r_k}\rangle + \|\boldsymbol{q}_{\boldsymbol{a}_k,r_k}\|^2\right)} \tag{31}
$$

$$
= 2\sqrt{c}\frac{-\left(1 - 2c\langle \boldsymbol{x}, \boldsymbol{q}_{\boldsymbol{a}_k,r_k}\rangle + c\|\boldsymbol{x}\|^2\right)\langle \boldsymbol{q}_{\boldsymbol{a}_k,r_k}, [\boldsymbol{a}_k]\rangle + \left(1 - c\|\boldsymbol{q}_{\boldsymbol{a}_k,r_k}\|^2\right)\langle \boldsymbol{x}, [\boldsymbol{a}_k]\rangle}{1 - c\|\boldsymbol{q}_{\boldsymbol{a}_k,r_k}\|^2 - c\|\boldsymbol{x}\|^2 + c^2\|\boldsymbol{x}\|^2\|\boldsymbol{q}_{\boldsymbol{a}_k,r_k}\|^2} \tag{32}
$$

$$
= \frac{2}{1 - c\|\boldsymbol{x}\|^2}\left(-\frac{\sqrt{c}\left(1 - 2c\langle \boldsymbol{x}, \boldsymbol{q}_{\boldsymbol{a}_k,r_k}\rangle + c\|\boldsymbol{x}\|^2\right)\langle \boldsymbol{q}_{\boldsymbol{a}_k,r_k}, [\boldsymbol{a}_k]\rangle}{1 - c\|\boldsymbol{q}_{\boldsymbol{a}_k,r_k}\|^2} + \sqrt{c}\langle \boldsymbol{x}, [\boldsymbol{a}_k]\rangle\right). \tag{33}
$$

With Equations 5 and 17, the term in the outer brackets in Equation 33 can be further expanded into the form using $r_k$ and $\boldsymbol{z}_k$ described in Section 3.1:

$$
-\frac{\sqrt{c}\left(1 - 2c\langle \boldsymbol{x}, \boldsymbol{q}_{\boldsymbol{a}_k,r_k}\rangle + c\|\boldsymbol{x}\|^2\right)\langle \boldsymbol{q}_{\boldsymbol{a}_k,r_k}, [\boldsymbol{a}_k]\rangle}{1 - c\|\boldsymbol{q}_{\boldsymbol{a}_k,r_k}\|^2} + \sqrt{c}\langle \boldsymbol{x}, [\boldsymbol{a}_k]\rangle
$$

$$
= -\frac{\left(1 - 2\sqrt{c}\tanh\left(\sqrt{c}\,r_k\right)\langle \boldsymbol{x}, [\boldsymbol{z}_k]\rangle + c\|\boldsymbol{x}\|^2\right)\tanh\left(\sqrt{c}\,r_k\right)}{1 - \tanh^2\left(\sqrt{c}\,r_k\right)} + \sqrt{c}\langle \boldsymbol{x}, [\boldsymbol{z}_k]\rangle \tag{34}
$$

$$
= -\left(1 + c\|\boldsymbol{x}\|^2\right)\sinh\left(\sqrt{c}\,r_k\right)\cosh\left(\sqrt{c}\,r_k\right) + \sqrt{c}\langle \boldsymbol{x}, [\boldsymbol{z}_k]\rangle\left(1 + 2\sinh^2\left(\sqrt{c}\,r_k\right)\right) \tag{35}
$$

$$
= -\frac{1 + c\|\boldsymbol{x}\|^2}{2}\sinh\left(2\sqrt{c}\,r_k\right) + \sqrt{c}\langle \boldsymbol{x}, [\boldsymbol{z}_k]\rangle\cosh\left(2\sqrt{c}\,r_k\right). \tag{36}
$$

In addition, we can also expand the term outside the $\sinh^{-1}$ function in Equation 29 using Equations 5 and 17 as follows:

$$
\frac{\lambda^c_{\boldsymbol{q}_{\boldsymbol{a}_k,r_k}}\|\boldsymbol{a}_k\|}{\sqrt{c}} = \frac{2\|\boldsymbol{a}_k\|}{\sqrt{c}\left(1 - c\|\boldsymbol{q}_{\boldsymbol{a}_k,r_k}\|^2\right)} = \frac{2\left\|\operatorname{sech}^2\left(\sqrt{c}\,r_k\right)\boldsymbol{z}_k\right\|}{\sqrt{c}\left(1 - \tanh^2\left(\sqrt{c}\,r_k\right)\right)} = \frac{2\|\boldsymbol{z}_k\|}{\sqrt{c}}. \tag{37}
$$

Combining Equations 29, 33, 36, and 37, we finally conclude the proof through the following:

$$
v_k(\boldsymbol{x}) = \frac{2\|\boldsymbol{z}_k\|}{\sqrt{c}}\sinh^{-1}\left(\frac{2\sqrt{c}\langle \boldsymbol{x}, [\boldsymbol{z}_k]\rangle}{1 - c\|\boldsymbol{x}\|^2}\cosh\left(2\sqrt{c}\,r_k\right) - \frac{1 + c\|\boldsymbol{x}\|^2}{1 - c\|\boldsymbol{x}\|^2}\sinh\left(2\sqrt{c}\,r_k\right)\right) \tag{38}
$$

$$
= \frac{2\|\boldsymbol{z}_k\|}{\sqrt{c}}\sinh^{-1}\left(\lambda^c_{\boldsymbol{x}}\langle \sqrt{c}\,\boldsymbol{x}, [\boldsymbol{z}_k]\rangle\cosh\left(2\sqrt{c}\,r_k\right) - (\lambda^c_{\boldsymbol{x}} - 1)\sinh\left(2\sqrt{c}\,r_k\right)\right). \tag{39}
$$

$\square$

## D.2 Convergence proof of unidirectional Poincaré MLR to Euclidean MLR

*Proof.* For the intended proof, we first introduce the following proposition:

**Proposition 1.** *For $x \neq 0$, $\sinh(x)$ over $x$ converges to 1 in the limit $x \to 0$:*

$$
\lim_{x \to 0}\frac{\sinh(x)}{x} = 1. \tag{40}
$$

*Proof.* The result can be obtained based on the definition of the differentiation of a scalar function:

$$
\lim_{x \to 0}\frac{\sinh(x)}{x} = \lim_{x \to 0}\frac{e^x - e^{-x}}{2x} = \frac{1}{2}\lim_{x \to 0}\left(\frac{e^x - 1}{x} + \frac{e^{-x} - 1}{-x}\right) \tag{41}
$$

$$
= \lim_{x \to 0}\frac{e^x - e^0}{x} = \left.\frac{de^x}{dx}\right|_{x=0} = 1. \tag{42}
$$

$\square$

From Proposition 1, we derive the following two propositions.

**Proposition 2.** *For $t \in \mathbb{R}, x \neq 0$, $\sinh(tx)$ over $x$ converges to $t$ in the limit $x \to 0$:*

$$\lim_{x \to 0} \frac{\sinh(tx)}{x} = t. \tag{43}$$

*Proof.* We divide this proof into two cases:

$$\lim_{x \to 0} \frac{\sinh(tx)}{x} = \begin{cases} 0 = t & (t = 0) \\ t \lim_{tx \to 0} \dfrac{\sinh(tx)}{tx} = t & (t \neq 0, \text{ Proposition 1}) \end{cases}. \tag{44}$$

$\square$

**Proposition 3.** *For $t \in \mathbb{R}, x \neq 0$, $\sinh^{-1}(tx)$ over $x$ converges to $t$ in the limit $x \to 0$:*

$$\lim_{x \to 0} \frac{\sinh^{-1}(tx)}{x} = t. \tag{45}$$

*Proof.* We can directly utilize Proposition 1 as follows:

$$\lim_{x \to 0} \frac{\sinh^{-1}(tx)}{x} = \lim_{s \to 0} \frac{ts}{\sinh(s)} \qquad (s = \sinh^{-1}(tx)) \tag{46}$$

$$= t \lim_{s \to 0} \left( \frac{\sinh(s)}{s} \right)^{-1} = t \qquad \text{(Proposition 1)}. \tag{47}$$

$\square$

With Propositions 2 and 3, we can now take the limit of Equation 6 as follows:

$$\lim_{c \to 0} v_k(\boldsymbol{x})$$
$$= \lim_{c \to 0} \frac{2\|\boldsymbol{z}_k\|}{\sqrt{c}} \sinh^{-1} \left( \frac{2\sqrt{c}\langle \boldsymbol{x}, [\boldsymbol{z}_k] \rangle}{1 - c\|\boldsymbol{x}\|^2} \cosh\left(2\sqrt{c}\,r_k\right) - \frac{1 + c\|\boldsymbol{x}\|^2}{1 - c\|\boldsymbol{x}\|^2} \sinh\left(2\sqrt{c}\,r_k\right) \right) \tag{48}$$

$$= \lim_{c \to 0} \frac{2\|\boldsymbol{z}_k\|}{\sqrt{c}} \sinh^{-1} \left( \sqrt{c} \left( \frac{2\langle \boldsymbol{x}, [\boldsymbol{z}_k] \rangle}{1 - c\|\boldsymbol{x}\|^2} \cosh\left(2\sqrt{c}\,r_k\right) - \frac{1 + c\|\boldsymbol{x}\|^2}{1 - c\|\boldsymbol{x}\|^2} \frac{\sinh\left(2\sqrt{c}\,r_k\right)}{\sqrt{c}} \right) \right) \tag{49}$$

$$= 2\|\boldsymbol{z}_k\| \left(2\langle \boldsymbol{x}, [\boldsymbol{z}_k] \rangle - 2r_k\right) = 4\left(\langle \boldsymbol{x}, \boldsymbol{z}_k \rangle - r_k \|\boldsymbol{z}_k\|\right). \tag{50}$$

Moreover, with Equation 5, we can confirm that $\boldsymbol{z}_k$ matches $\boldsymbol{a}_k$ in the limit $c \to 0$:

$$\lim_{c \to 0} \boldsymbol{a}_k = \lim_{c \to 0} \text{sech}^2\left(\sqrt{c}\,r_k\right) \boldsymbol{z}_k = \lim_{c \to 0} \frac{1}{\cosh^2\left(\sqrt{c}\,r_k\right)} \boldsymbol{z}_k = \boldsymbol{z}_k. \tag{51}$$

Combining it with Equations 2 and 50, we finally conclude the proof as follows:

$$\lim_{c \to 0} v_k(\boldsymbol{x}) = 4\left(\langle \boldsymbol{x}, \boldsymbol{a}_k \rangle - r_k \|\boldsymbol{a}_k\|\right) = 4\left(\langle \boldsymbol{a}_k, \boldsymbol{x} \rangle - b_k\right), \quad \text{where } b_k := r_k \|\boldsymbol{a}_k\|. \tag{52}$$

Here, the factor 4 is derived from the squared conformal factor $(\lambda_{\boldsymbol{x}}^0)^2$ degenerating into a constant value. This corresponds to the fact that the Poincaré ball model $\mathbb{B}_c^n$ converges to the Euclidean space $\mathbb{R}^n$ in the limit $c \to 0$ except for the same multiplier $\lim_{c \to 0}(\lambda_{\boldsymbol{x}}^c)^2 = 4$ owing to its metric tensor. $\square$

### D.3    PROOF OF THE PROPERTIES OF OUTPUT COORDINATES OF POINCARÉ FC LAYER

*Proof.* To check the properties of the Poincaré FC layer described in Section 3.2, we first clarify the Poincaré hyperplane containing the origin and orthogonal to the $k$-th axis in $\mathbb{B}_c^m$. The $k$-th axis is a geodesic passing through the origin and any point on it except the origin has a non-zero element in only the $k$-th coordinates. Therefore, an arbitrary point $\boldsymbol{x} \in \mathbb{B}_c^m$ along the $k$-th axis can be written as follows:

$$\boldsymbol{x} = r\boldsymbol{e}^{(k)}, \quad \text{where } \boldsymbol{e}^{(k)} = (\delta_{ik})_{i=1}^m, \ r \in \left(-\frac{1}{\sqrt{c}}, \frac{1}{\sqrt{c}}\right) \subset \mathbb{R}, \tag{53}$$

which is as intuitive as in a Euclidean space. Specifically, $r = 0$ represents the origin.

We can then easily describe the intended Poincaré hyperplane as follows:

**Definition 1.** *(Poincaré hyperplane containing the origin and orthogonal to the $k$-th axis)*

$$\bar{H}^c_{\boldsymbol{e}^{(k)},0} = \{\boldsymbol{x} = (x_1, x_2, \ldots, x_m)^\top \in \mathbb{B}^m_c \mid \langle \boldsymbol{e}^{(k)}, \boldsymbol{x} \rangle = x_k = 0\}, \tag{54}$$

which is also intuitively obtained.

With Definition 1, the preparation for constructing $\boldsymbol{y}$ in Equation 7 is complete.

**Derivation of $\boldsymbol{y}$.** Let $\boldsymbol{x} \in \mathbb{B}^n_c$ and $\boldsymbol{y} = (y_1, y_2, \ldots, y_m)^\top \in \mathbb{B}^m_c$ be the input and output of the Poincaré FC layer, respectively. Below, we start the proof with the score functions $v_k(\boldsymbol{x})$ for $\forall k = \{1, 2, \ldots, m\}$ already obtained in the same way as in Equation 6.

To endow $\boldsymbol{y}$ the properties described in Section 3.2, *i.e.*, the signed distance from $\boldsymbol{y}$ to each Poincaré hyperplane containing the origin and orthogonal to the $k$-th axis is equal to $v_k(\boldsymbol{x})$, we generate a simultaneous equation for $\forall k$ as follows:

$$d_c\left(\boldsymbol{y}, \bar{H}^c_{\boldsymbol{e}^{(k)},0}\right) = v_k(\boldsymbol{x}). \tag{55}$$

With Equations 54 and 28 and the notion of the distance from a point to a Poincaré hyperplane described in Equation 23, these equations are expanded as follows:

$$\frac{1}{\sqrt{c}} \sinh^{-1}\left(\frac{2\sqrt{c}\, y_k}{1 - c\|\boldsymbol{y}\|^2}\right) = v_k(\boldsymbol{x}). \tag{56}$$

Therefore, we obtain the following notation of the coordinates:

$$y_k = \frac{1 - c\|\boldsymbol{y}\|^2}{2\sqrt{c}} \sinh\left(\sqrt{c}\, v_k(\boldsymbol{x})\right), \quad \forall k. \tag{57}$$

When considering the Euclidean norm of $\boldsymbol{y}$ using Equation 57, the equation for $\|\boldsymbol{y}\|$ can be derived as follows:

$$\|\boldsymbol{y}\| = \frac{1 - c\|\boldsymbol{y}\|^2}{2\sqrt{c}} \sqrt{\sum_{k=1}^{m} \sinh^2\left(\sqrt{c}\, v_k(\boldsymbol{x})\right)}. \tag{58}$$

This can be succinctly rewritten as

$$\|\boldsymbol{y}\| = \frac{1 - c\|\boldsymbol{y}\|^2}{2} \|\boldsymbol{w}\|, \quad \text{where } \boldsymbol{w} = \left(\frac{1}{\sqrt{c}} \sinh\left(\sqrt{c}\, v_k(\boldsymbol{x})\right)\right)_{k=1}^{m}. \tag{59}$$

By solving this quadratic equation, the closed form of $\|\boldsymbol{y}\|$ is obtained through the following:

$$\|\boldsymbol{y}\| = -\frac{1}{c\|\boldsymbol{w}\|} + \sqrt{\frac{1}{c^2\|\boldsymbol{w}\|^2} + \frac{1}{c}}. \tag{60}$$

Substituting Equations 59 and 60 for Equation 57 leads to Equation 7 in the notation of the coordinates:

$$y_k = \frac{\sqrt{1 + c\|\boldsymbol{w}\|^2} - 1}{c\|\boldsymbol{w}\|^2} w_k = \frac{w_k}{1 + \sqrt{1 + c\|\boldsymbol{w}\|^2}}, \quad \forall k. \tag{61}$$

**Confirmation of the existence of $\boldsymbol{y}$.** Finally, we conclude the proof by checking that $\boldsymbol{y}$ is always within the domain of the Poincaré ball $\mathbb{B}^m_c = \{\boldsymbol{y} \in \mathbb{R}^m \mid c\|\boldsymbol{y}\|^2 < 1\}$:

$$1 - c\|\boldsymbol{y}\|^2 = \frac{2\left(\sqrt{1 + c\|\boldsymbol{w}\|^2} - 1\right)}{c\|\boldsymbol{w}\|^2} > 0. \tag{62}$$

$\square$

D.4 PROOF OF THE PROPERTIES OF POINCARÉ $\boldsymbol{\beta}$-SPLIT AND $\boldsymbol{\beta}$-CONCATENATION

In this section, we prove the properties of the Poincaré $\beta$-split and the Poincaré $\beta$-concatenation described in Section 3.3. The Poincaré ball model is different from Euclidean neural networks, on the simple calculation of the expected value and the variance of a particular value related to a feature vector or weight matrix owing to the linearity in their operations. In the Poincaré ball model, calculating such values without any postulate for the probabilistic distribution that the feature gyrovectors or tangent vectors follow is difficult owing to the nonlinear transformations in the exponential and logarithmic maps. Thus, we first make the following naive assumption:

**Assumption 1.** *Each coordinate of an $n$-dimensional tangent vector in $\mathcal{T}_0 \mathbb{B}_c^n$ follows a normal distribution centered at zero with a certain variance $\frac{\sigma_n^2}{c}$.*

The reasons why we assume the distribution on the tangent space rather than on the Poincaré ball model itself are as follows:

1. It is improper to assume a continuous and smooth distribution onto the space with an upper-bounded radius because there must be no probability density on or outside the boundary. The rough idea of discontinuing such probabilities outside the domain of the Poincaré ball and discretely taking only the inside into account seems to lack rationality.

2. One simple way to avoid the above issue is to apply a uniform distribution from zero to the ball radius based on the norm of the gyrovector. However, there is no guarantee that such constancy in the distribution can be realized on a complexly curved geometric structure of the Poincaré ball model.

3. Conversely, a tangent space is a linear space that is attached to the manifold and can be treated as an ordinary vector space.

4. The Poincaré ball model is conformal to the Euclidean space, *i.e.*, preserving the same angles, and at the origin, the gyrovectors having the same norms are projected onto the tangent vectors which also have the same norms with their angles unchanged.

5. In Euclidean neural networks, the normal distribution is one of the most popularly considered priors. The multivariate normal distribution is occasionally approximated as an independent and identically distributed distribution for easier calculation.

Because the Poincaré $\beta$-split and the Poincaré $\beta$-concatenation are inverse functions to each other, it is sufficient to prove the properties of either one of these operations. Here, we show a proof for the Poincaré $\beta$-concatenation. Recalling that $\beta_n = \mathrm{B}(\frac{n}{2}, \frac{1}{2})$ and considering the following:

**Poincaré $\boldsymbol{\beta}$-concatenation.** The input gyrovectors $\{\boldsymbol{x}_i \in \mathbb{B}_c^{n_i}\}_{i=1}^N$ are first scaled by certain coefficients and concatenated in the tangent space, and then projected back to the Poincaré ball as follows:

$$\boldsymbol{x}_i \mapsto \boldsymbol{v}_i = \log_{\boldsymbol{0}}^c(\boldsymbol{x}_i) \in \mathcal{T}_{\boldsymbol{0}} \mathbb{B}_c^{n_i}, \quad \boldsymbol{v} := \left( \frac{\beta_n}{\beta_{n_1}} \boldsymbol{v}_1^\top, \ldots, \frac{\beta_n}{\beta_{n_N}} \boldsymbol{v}_N^\top \right)^\top \mapsto \boldsymbol{y} = \exp_{\boldsymbol{0}}^c(\boldsymbol{v}) \in \mathbb{B}_c^n. \quad (63)$$

*Proof.* At first, we consider the expected value of the norm of each tangent vector $\boldsymbol{v}_i$, which is the target of the Poincaré $\beta$-concatenation. Because the value $t_i := \frac{c\|\boldsymbol{v}_i\|^2}{\sigma_{n_i}^2}$ follows a $\chi^2$ distribution based on Assumption 1, the expected value of $\|\boldsymbol{v}_i\|$ can be obtained as follows:

$$E[\|\boldsymbol{v}_i\|] = \frac{1}{2^{\frac{n_i}{2}} \Gamma\left(\frac{n_i}{2}\right)} \int_0^\infty \|\boldsymbol{v}_i\| e^{-\frac{t_i}{2}} t_i^{\frac{n_i}{2}-1} dt_i \tag{64}$$

$$= \frac{\sigma_{n_i}}{2^{\frac{n_i}{2}} \Gamma\left(\frac{n_i}{2}\right) \sqrt{c}} \int_0^\infty e^{-\frac{t_i}{2}} t_i^{\frac{n_i-1}{2}} dt_i \tag{65}$$

$$= \frac{2^{\frac{n_i+1}{2}} \Gamma\left(\frac{n_i+1}{2}\right)}{2^{\frac{n_i}{2}} \Gamma\left(\frac{n_i}{2}\right)} \frac{\sigma_{n_i}}{\sqrt{c}} \tag{66}$$

$$= \sqrt{\frac{2\pi}{c}} \frac{\sigma_{n_i}}{\mathrm{B}\left(\frac{n_i}{2}, \frac{1}{2}\right)} \tag{67}$$

$$= \sqrt{\frac{2\pi}{c}} \frac{\sigma_{n_i}}{\beta_{n_i}}. \tag{68}$$

Therefore, when the norm of each input tangent vector $\boldsymbol{v}_i$ is kept the same by the former part of neural networks before applying this operation, the standard deviation $\sigma_{n_i}$ must be expressed as follows:

$$\sigma_{n_i} = C\beta_{n_i}, \quad \text{where } C = const. \tag{69}$$

In addition, using Equation 63, the squared norm of the Poincaré $\beta$-concatenated tangent vector $\boldsymbol{v}$ is obtained as follows:

$$\|\boldsymbol{v}\|^2 = \sum_{i=1}^{N} \left(\frac{\beta_n}{\beta_{n_i}}\right)^2 \|\boldsymbol{v}_i\|^2 = \sum_{i=1}^{N} \frac{\beta_n^2}{c} \frac{c\|\boldsymbol{v}_i\|^2}{\sigma_{n_i}^2} C^2 = \frac{\beta_n^2 C^2}{c} \sum_{i=1}^{N} t_i. \tag{70}$$

This leads the value $t := \frac{c\|\boldsymbol{v}\|^2}{\sigma_n^2}$, where $\sigma_n = C\beta_n$, which is expressed as follows:

$$t = \sum_{i=1}^{N} t_i. \tag{71}$$

Here, $t$ also follows a $\chi^2$ distribution, and the expected value of the norm of $\boldsymbol{v}$ is obtained as follows:

$$E[\|\boldsymbol{v}\|] = \frac{1}{2^{\frac{n}{2}}\Gamma\left(\frac{n}{2}\right)} \int_0^\infty \|\boldsymbol{v}\| e^{-\frac{t}{2}} t^{\frac{n}{2}-1} dt = \sqrt{\frac{2\pi}{c}} \frac{\sigma_n}{\beta_n} = \sqrt{\frac{2\pi}{c}} C, \tag{72}$$

which is the same as the norms of the input tangent vectors. This indicates that each coordinate of $\boldsymbol{v}$ follows a normal distribution centered at zero with a variance $\frac{\sigma_n^2}{c}$, satisfying the Assumption 1.

Based on the results above, the expected value of the norm of each input gyrovector $\boldsymbol{x}_i$ is expressed by the following:

$$E[\|\boldsymbol{x}_i\|] = \int_0^\infty \|\boldsymbol{x}_i\| \frac{1}{2^{\frac{n_i}{2}}\Gamma\left(\frac{n_i}{2}\right)} e^{-\frac{t_i}{2}} t_i^{\frac{n_i}{2}-1} dt_i \tag{73}$$

$$= \frac{1}{2^{\frac{n_i}{2}}\Gamma\left(\frac{n_i}{2}\right)} \int_0^\infty \frac{1}{\sqrt{c}} \tanh\left(\sqrt{c}\|\boldsymbol{v}_i\|\right) e^{-\frac{t_i}{2}} t_i^{\frac{n_i}{2}-1} dt_i \tag{74}$$

$$= \frac{1}{2^{\frac{n_i}{2}}\Gamma\left(\frac{n_i}{2}\right)\sqrt{c}} \int_0^\infty \tanh\left(\sigma_{n_i}\sqrt{t_i}\right) e^{-\frac{t_i}{2}} t_i^{\frac{n_i}{2}-1} dt_i \tag{75}$$

$$= \frac{1}{2^{\frac{n_i}{2}}\Gamma\left(\frac{n_i}{2}\right)\sqrt{c}} \int_0^\infty \sum_{j=1}^\infty \frac{2^{2j}\left(2^{2j}-1\right)B_{2j}\left(\sigma_{n_i}\sqrt{t_i}\right)^{2j-1}}{(2j)!} e^{-\frac{t_i}{2}} t_i^{\frac{n_i}{2}-1} dt_i \tag{76}$$

$$= \frac{1}{2^{\frac{n_i}{2}}\Gamma\left(\frac{n_i}{2}\right)\sqrt{c}} \sum_{j=1}^\infty \frac{2^{2j}\left(2^{2j}-1\right)B_{2j}\sigma_{n_i}^{2j-1}}{(2j)!} \int_0^\infty e^{-\frac{t_i}{2}} t_i^{\frac{n_i-3}{2}+j} dt_i \tag{77}$$

$$= \frac{1}{2^{\frac{n_i}{2}}\Gamma\left(\frac{n_i}{2}\right)\sqrt{c}} \sum_{j=1}^\infty \frac{2^{2j}\left(2^{2j}-1\right)B_{2j}\sigma_{n_i}^{2j-1}}{(2j)!} 2^{j+\frac{n_i-1}{2}}\Gamma\left(j+\frac{n_i-1}{2}\right) \tag{78}$$

$$= \frac{1}{\sqrt{c}} \sum_{j=1}^\infty \frac{2^{2j}\left(2^{2j}-1\right)B_{2j}}{(2j)!} \left(\sqrt{2\pi}C\right)^{2j-1} \frac{\Gamma\left(\frac{n_i}{2}\right)^{2j-2}}{\Gamma\left(\frac{n_i+1}{2}\right)^{2j-1}}\Gamma\left(j+\frac{n_i-1}{2}\right). \tag{79}$$

Note that, for the calculation between Equations 75 and 76, we utilize the Taylor series expansion of $\tanh$ for a real value. Furthermore, considering the Laurent series expansion at infinity, we can obtain the following expressions:

$$\Gamma\left(j+\frac{n_i-1}{2}\right) = (2e)^{-\frac{n_i}{2}} n_i^{j+\frac{n_i}{2}} \left(\frac{2^{\frac{3}{2}-j}\sqrt{\pi}}{n_i} + O\left(\frac{1}{n_i^2}\right)\right), \tag{80}$$

$$\frac{\Gamma\left(\frac{n_i+1}{2}\right)}{\Gamma\left(\frac{n_i}{2}\right)^2} = (2e)^{\frac{n_i}{2}} n_i^{2-\frac{n_i}{2}} \left(\frac{1}{2^{\frac{3}{2}}\sqrt{\pi}n_i} + O\left(\frac{1}{n_i^2}\right)\right). \tag{81}$$

Therefore, in the general cases in which $n_i \gg 1$, we can obtain the following approximation:

$$\frac{\Gamma\left(\frac{n_i}{2}\right)^{2j-2}}{\Gamma\left(\frac{n_i+1}{2}\right)^{2j-1}}\Gamma\left(j+\frac{n_i-1}{2}\right) = \Gamma\left(j+\frac{n_i-1}{2}\right)\frac{\Gamma\left(\frac{n_i+1}{2}\right)}{\Gamma\left(\frac{n_i}{2}\right)^2}\left(\frac{\Gamma\left(\frac{n_i}{2}\right)}{\Gamma\left(\frac{n_i+1}{2}\right)}\right)^{2j} \tag{82}$$

$$\simeq (2e)^{\frac{n_i}{2}-\frac{n_i}{2}}\frac{2^{\frac{3}{2}-j}\sqrt{\pi}}{2^{\frac{3}{2}}\sqrt{\pi}}n_i^{j+\frac{n_i}{2}-\frac{n_i}{2}+2-2}\left(\frac{\Gamma\left(\frac{n_i}{2}\right)}{\Gamma\left(\frac{n_i+1}{2}\right)}\right)^{2j} \tag{83}$$

$$= 2^{-j}n_i^j\left(\frac{\Gamma\left(\frac{n_i}{2}\right)}{\Gamma\left(\frac{n_i+1}{2}\right)}\right)^{2j} \tag{84}$$

$$\simeq 2^{-j}n_i^j\left(\frac{\sqrt{\pi}\,(2e)^{-\frac{n_i}{2}}\,n_i^{\frac{n_i-1}{2}}}{\sqrt{\pi}\,(2e)^{-\frac{n_i+1}{2}}\,(n_i+1)^{\frac{n_i}{2}}}\right)^{2j} \tag{85}$$

$$= 2^{-j}n_i^j\left((2e)^{\frac{1}{2}}\frac{n_i^{\frac{n_i-1}{2}}}{(n_i+1)^{\frac{n_i}{2}}}\right)^{2j} \tag{86}$$

$$= 2^{-j}n_i^j(2e)^j\frac{n_i^{j(n_i-1)}}{(n_i+1)^{jn_i}} \tag{87}$$

$$= e^j\left(\frac{n_i}{n_i+1}\right)^{n_i j} \tag{88}$$

$$\simeq e^j e^{-j} \tag{89}$$

$$= 1. \tag{90}$$

Note that, for the calculation between Equations 84 and 85, we utilize Stirling's approximation, *i.e.*, $\Gamma(z) \simeq \sqrt{\frac{2\pi}{z}}\left(\frac{z}{e}\right)^z$. In addition, we utilize the definition of Napier's constant for the approximation between Equations 88 and 89, *i.e.*, $\lim_{x\to\infty}(1+\frac{1}{x})^x = e$.

Combining Equations 79 and 90, the expected value of $\|\boldsymbol{x}_i\|$ can be approximately expressed by the following:

$$E[\|\boldsymbol{x}_i\|] \simeq \frac{1}{\sqrt{c}}\sum_{j=1}^{\infty}\frac{2^{2j}\left(2^{2j}-1\right)B_{2j}}{(2j)!}\left(\sqrt{2\pi}C\right)^{2j-1} \tag{91}$$

$$= \frac{1}{\sqrt{c}}\tanh\left(\sqrt{2\pi}C\right) \tag{92}$$

$$= \frac{1}{\sqrt{c}}\tanh\left(\sqrt{c}\,E[\|\boldsymbol{v}_i\|]\right). \tag{93}$$

In the same way, the expected value of the Poincaré $\beta$-concatenated gyrovector $\boldsymbol{x}$ is obtained by the following:

$$E[\|\boldsymbol{x}\|] \simeq \frac{1}{\sqrt{c}}\tanh\left(\sqrt{2\pi}C\right) \tag{94}$$

$$= \frac{1}{\sqrt{c}}\tanh\left(\sqrt{c}\,E[\|\boldsymbol{v}\|]\right), \tag{95}$$

which concludes the proof. $\qquad\square$

### D.5 THE MÖBIUS GYROMIDPOINT AND THE EINSTEIN GYROMIDPOINT

**Einstein gyromidpoint.** In the Beltrami-Klein model, the midpoint $\bar{n} \in \mathbb{K}_c^n$ among $\{n_i \in \mathbb{K}_c^n\}_{i=1}^N$ and the non-negative scalar weights $\{\nu_i \in \mathbb{R}_+\}_{i=1}^N$ is given as follows:

$$\bar{n} = \frac{\sum_{i=1}^N \nu_i \gamma_i n_i}{\sum_{i=1}^N \nu_i \gamma_i}, \quad \text{where } \gamma_i = \frac{1}{\sqrt{1 - c\|n_i\|^2}}. \tag{96}$$

This operation is called the Einstein gyromidpoint (Ungar, 2009).

Based on the above, we prove the equivalence of the Möbius gyromidpoint and Einstein gyromidpoint.

*Proof.* Let the points $\{b_i \in \mathbb{B}_c^n\}_{i=1}^N$ correspond to $\{n_i \in \mathbb{K}_c^n\}_{i=1}^N$, respectively, *i.e.*, $b_i$ is a projection of $n_i$ to the Poincaré ball model using Equation 11. From Equations 12 and 96, we obtain the following:

$$\gamma_i = \frac{1}{\sqrt{1 - c\|n_i\|^2}} = \frac{1 + c\|b_i\|^2}{1 - c\|b_i\|^2}. \tag{97}$$

Substituting Equations 12 and 97 for Equation 96 leads to the representation of the Einstein midpoint using the coordinates in the Poincaré ball model:

$$\bar{n} = \frac{\sum_{i=1}^N \nu_i \frac{2b_i}{1 - c\|b_i\|^2}}{\sum_{i=1}^N \nu_i \frac{1 + c\|b_i\|^2}{1 - c\|b_i\|^2}} = \frac{\sum_{i=1}^N \nu_i \lambda_{b_i}^c b_i}{\sum_{i=1}^N \nu_i \left(\lambda_{b_i}^c - 1\right)}. \tag{98}$$

Therefore, the point $\bar{b} \in \mathbb{B}_c^n$, which is a projection of $\bar{n}$ to the Poincaré ball model using Equation 11, is expressed in the following manner:

$$\bar{b} = \frac{\underline{b}}{1 + \sqrt{1 - c\|\underline{b}\|^2}} = \frac{1}{2} \otimes_c \underline{b}, \quad \text{where } \underline{b} = \bar{n} = \frac{\sum_{i=1}^N \nu_i \lambda_{b_i}^c b_i}{\sum_{i=1}^N \nu_i \left(\lambda_{b_i}^c - 1\right)}. \tag{99}$$

This concludes the proof. $\square$

### D.6 MÖBIUS GYROMIDPOINT AND CENTROID OF SQUARED LORENTZIAN DISTANCE

**Weighted centroid in the hyperboloid model** (Law et al., 2019). With a Lorentzian norm $\|\|x\|_{\mathcal{L}}\| = \sqrt{|\langle x, x \rangle_{\mathcal{L}}|} = \sqrt{|\|x\|_{\mathcal{L}}^2|}$ for $x \in \mathbb{R}_1^{n+1}$, the center of mass $\bar{h} \in \mathbb{H}_c^n$ among $\{h_i = (z_i, k_i^\top)^\top \in \mathbb{H}_c^n\}_{i=1}^N$ and the non-negative scalar weights $\{\nu_i \in \mathbb{R}_+\}_{i=1}^N$ is given as follows:

$$\bar{h} = \frac{\underline{h}}{\sqrt{c}|\|\underline{h}\|_{\mathcal{L}}|}, \quad \text{where } \underline{h} = \sum_{i=1}^N \nu_i h_i. \tag{100}$$

This is based on the minimization problem of the weighted sum of squared Lorentzian distances expressed as follows:

$$\bar{h} = \arg\min_{\tilde{h}} \sum_{i=1}^N \nu_i \|h_i - \tilde{h}\|_{\mathcal{L}}^2. \tag{101}$$

In the following, we prove the equivalence of the Möbius gyromidpoint and the weighted centroid in the hyperboloid model.

*Proof.* Expanding Equation 100 with the coordinates, we obtain the following:

$$\bar{\boldsymbol{h}} = \frac{1}{\sqrt{c}} \frac{\left( \sum_{i=1}^{N} \nu_i z_i, \sum_{i=1}^{N} \nu_i \boldsymbol{k}_i^{\top} \right)^{\top}}{\sqrt{\left( \sum_{i=1}^{N} \nu_i z_i \right)^2 - \left\| \sum_{i=1}^{N} \nu_i \boldsymbol{k}_i \right\|^2}}. \tag{102}$$

The point $\bar{\boldsymbol{b}} \in \mathbb{B}_c^n$, which is a projection of $\bar{\boldsymbol{h}}$ to the Poincaré ball model using Equation 9, is expressed in the following manner:

$$\bar{\boldsymbol{b}} = \frac{1}{\sqrt{c}} \frac{\sum_{i=1}^{N} \nu_i \boldsymbol{k}_i}{\sqrt{\left( \sum_{i=1}^{N} \nu_i z_i \right)^2 - \left\| \sum_{i=1}^{N} \nu_i \boldsymbol{k}_i \right\|^2} + \sum_{i=1}^{N} \nu_i z_i}. \tag{103}$$

Dividing both the numerator and denominator by $\sum_i \nu_i z_i$, this can be rewritten as follows:

$$\bar{\boldsymbol{b}} = \frac{\underline{\boldsymbol{b}}}{1 + \sqrt{1 - c\|\underline{\boldsymbol{b}}\|^2}} = \frac{1}{2} \otimes_c \underline{\boldsymbol{b}}, \quad \text{where } \underline{\boldsymbol{b}} := \frac{1}{\sqrt{c}} \frac{\sum_{i=1}^{N} \nu_i \boldsymbol{k}_i}{\sum_{i=1}^{N} \nu_i z_i}. \tag{104}$$

Next, considering the points $\{\boldsymbol{b}_i \in \mathbb{B}_c^n\}_{i=1}^N$, which also correspond to $\{\boldsymbol{h}_i\}_{i=1}^N$, respectively, we can transform the expression of $\underline{\boldsymbol{b}}$ into an expression with only the coordinates in the Poincaré ball model:

$$\underline{\boldsymbol{b}} = 2 \frac{\sum_{i=1}^{N} \nu_i \frac{\boldsymbol{b}_i}{1 - c\|\boldsymbol{b}_i\|^2}}{\sum_{i=1}^{N} \nu_i \frac{1 + c\|\boldsymbol{b}_i\|^2}{1 - c\|\boldsymbol{b}_i\|^2}} = \frac{\sum_{i=1}^{N} \nu_i \lambda_{\boldsymbol{b}_i}^c \boldsymbol{b}_i}{\sum_{i=1}^{N} \nu_i \left( \lambda_{\boldsymbol{b}_i}^c - 1 \right)}. \tag{105}$$

This concludes the proof. □

## D.7 Möbius gyromidpoint as a solution of the minimization problem

The discovery of the equivalence between the weighted centroid in the hyperboloid model and the Möbius gyromidpoint enables us to discuss what the Möbius gyromidpoint is a minimizer of. In the following, we prove that the Möbius gyromidpoint can be regarded as a minimizer of the weighted sum of calibrated squared gyrometrics.

**Theorem 2.** *The Möbius gyromidpoint is a solution of the minimization problem of the weighted sum of calibrated squared gyrometrics, which is expressed as follows:*

$$\bar{\boldsymbol{b}} = \arg\min_{\tilde{\boldsymbol{b}}} \sum_{i=1}^{N} \nu_i \, \lambda_{\ominus_c \tilde{\boldsymbol{b}} \oplus_c \boldsymbol{b}_i}^c \| \ominus_c \tilde{\boldsymbol{b}} \oplus_c \boldsymbol{b}_i \|^2. \tag{106}$$

Each $\| \ominus_c \bar{\boldsymbol{b}} \oplus_c \boldsymbol{b}_i \|$ indicates the norm of the respective gyrovector $\boldsymbol{b}_i$ viewed from the Möbius gyromidpoint $\bar{\boldsymbol{b}}$, which equals the gyrodistance of $\bar{\boldsymbol{b}}$ to $\boldsymbol{b}_i$ and is also called a gyrometric (Ungar, 2009). In addition, each $\lambda_{\ominus_c \bar{\boldsymbol{b}} \oplus_c \boldsymbol{b}_i}^c$ is a conformal factor of the metric tensor of the Poincaré ball model for such a gyrovector. Therefore, the minimization objective in Equation 106 can be interpreted as the weighted sum of squared gyrometrics, each of which is calibrated by a scaling factor at the respective point.

*Proof.* Let the point $\bar{\boldsymbol{b}} \in \mathbb{B}_c^n$ be a projection of the weighted centroid $\bar{\boldsymbol{h}} \in \mathbb{H}_c^n$. With Equation 101 and the notation of Equation 10, we obtain the following straightforward expression:

$$\bar{\boldsymbol{b}} = \arg\min_{\tilde{\boldsymbol{b}}} \sum_{i=1}^{N} \nu_i \|\boldsymbol{h}_i - \boldsymbol{h}(\tilde{\boldsymbol{b}})\|_{\mathcal{L}}^2. \tag{107}$$

Expanding Equation 107 with the coordinates, we obtain the following:

$$\bar{\boldsymbol{b}} = \arg\min_{\tilde{\boldsymbol{b}}} -2 \sum_{i=1}^{N} \nu_i \left( \frac{1}{c} - z_i z(\tilde{\boldsymbol{b}}) + \langle \boldsymbol{h}_i, \boldsymbol{h}(\tilde{\boldsymbol{b}}) \rangle \right) \tag{108}$$

$$= \arg\min_{\tilde{\boldsymbol{b}}} \sum_{i=1}^{N} \nu_i \left( -\frac{1}{c} + z_i z(\tilde{\boldsymbol{b}}) - \langle \boldsymbol{h}_i, \boldsymbol{h}(\tilde{\boldsymbol{b}}) \rangle \right). \tag{109}$$

Considering the points $\{\boldsymbol{b}_i \in \mathbb{B}_c^n\}_{i=1}^N$, which correspond to $\{\boldsymbol{h}_i\}_{i=1}^N$, respectively, we can transform Equation 109 into an expression with only the coordinates in the Poincaré ball model:

$$\bar{\boldsymbol{b}} = \arg\min_{\tilde{\boldsymbol{b}}} \sum_{i=1}^{N} \nu_i \left( -\frac{1}{c} + \frac{1}{c} \frac{1+c\|\boldsymbol{b}_i\|^2}{1-c\|\boldsymbol{b}_i\|^2} \frac{1+c\|\tilde{\boldsymbol{b}}\|^2}{1-c\|\tilde{\boldsymbol{b}}\|^2} - \langle \frac{2\boldsymbol{b}_i}{1-c\|\boldsymbol{b}_i\|^2}, \frac{2\tilde{\boldsymbol{b}}}{1-c\|\tilde{\boldsymbol{b}}\|^2} \rangle \right) \tag{110}$$

$$= \arg\min_{\tilde{\boldsymbol{b}}} \sum_{i=1}^{N} \frac{2\nu_i \|\boldsymbol{b}_i - \tilde{\boldsymbol{b}}\|^2}{(1-c\|\boldsymbol{b}_i\|^2)(1-c\|\tilde{\boldsymbol{b}}\|^2)} \tag{111}$$

$$= \arg\min_{\tilde{\boldsymbol{b}}} \sum_{i=1}^{N} \frac{2\nu_i \| \ominus_c \tilde{\boldsymbol{b}} \oplus_c \boldsymbol{b}_i \|^2}{1 - c\| \ominus_c \tilde{\boldsymbol{b}} \oplus_c \boldsymbol{b}_i \|^2} \tag{112}$$

$$= \arg\min_{\tilde{\boldsymbol{b}}} \sum_{i=1}^{N} \nu_i \lambda_{\ominus_c \tilde{\boldsymbol{b}} \oplus_c \boldsymbol{b}_i}^c \| \ominus_c \tilde{\boldsymbol{b}} \oplus_c \boldsymbol{b}_i \|^2. \tag{113}$$

This concludes the proof. $\square$

### D.8 Weight generalization of the Möbius gyromidpoint

As mentioned in Section 3.5, we extend the condition of the weights of the Möbius gyromidpoint to all real values $\{\nu_i \in \mathbb{R}\}_{i=1}^N$ by regarding a negative weight as an additive inverse operation, that is, regarding any pair $(\nu_i, \boldsymbol{b}_i)$ as $(|\nu_i|, \text{sign}(\nu_i)\boldsymbol{b}_i)$:

$$\frac{\sum_{i=1}^{N} |\nu_i| \lambda_{\text{sign}(\nu_i)\boldsymbol{b}_i}^c \text{sign}(\nu_i)\boldsymbol{b}_i}{\sum_{i=1}^{N} |\nu_i| \left( \lambda_{\text{sign}(\nu_i)\boldsymbol{b}_i}^c - 1 \right)} = \frac{\sum_{i=1}^{N} \nu_i \lambda_{\boldsymbol{b}_i}^c \boldsymbol{b}_i}{\sum_{i=1}^{N} |\nu_i| \left( \lambda_{\boldsymbol{b}_i}^c - 1 \right)}. \tag{114}$$

## E Implementation Details

### E.1 Parameter initialization

**Unidirectional Poincaré MLR.** When the dimensions of the input gyrovector is $n$, each element of the weight parameter $\boldsymbol{Z}$ is initialized by a normal distribution centered at zero with a standard deviation $n^{-\frac{1}{2}}$. The bias parameter $\boldsymbol{r}$ is initialized as a zero vector.

**Poincaré FC layer.** When the dimensions of the input gyrovector and the output gyrovector are $n$ and $m$, respectively, each element of the weight parameter $\boldsymbol{Z}$ is initialized by a normal distribution centered at zero with a standard deviation $(2nm)^{-\frac{1}{2}}$. The bias parameter $\boldsymbol{r}$ is initialized as a zero vector.

**Poincaré convolutional layer.**   When the dimensions of the input gyrovector and the output gyrovector are $n$ and $m$, respectively, and the total kernel size is $K$, each element of the weight parameter $\boldsymbol{Z}$ is initialized by a normal distribution centered at zero with a standard deviation $(2nKm)^{-\frac{1}{2}}$. The bias parameter $\boldsymbol{r}$ is initialized as a zero vector.

**Embedding on the Poincaré ball model.**   As mentioned by Ganea et al. (2018a), we confirmed the tendency of the parameters in the Poincaré ball model to adjust their angles at the first phase of the training before increasing their norms. In addition, we consider that, due to the exponentially growing distance metric of the hyperbolic space, the farther a gyrovector parameter is placed from the origin, the more costly it moves such a point to another point through the optimization. Therefore, the embedding parameters on the Poincaré ball model should be initialized with a particular small gain $\epsilon_E$, given as a hyperparameter, aiming to accelerate such an adjustment and make the later optimization smooth. We set the value $\epsilon_E$ to be $10^{-2}$ in the experiment in Section 4.3.

### E.2   HYPERPARAMETERS OF THE EXPERIMENT IN SECTION 4.2

**Optimization.**   We used the Riemannian Adam optimizer with $\beta_1 = 0.9$, $\beta_2 = 0.999$ and $\epsilon = 10^{-8}$ for both of the Euclidean and our hyperbolic architectures. The learning rate $\eta$ was set to $10^{-3}$.

### E.3   HYPERPARAMETERS OF THE EXPERIMENT IN SECTION 4.3

**Model architectures.**   Let $D$ be the dimension of the source and target token embeddings. Each model for the experiment in Section 4.3 has the encoder and decoder, both of which are composed of five convolutional layers with a kernel size of three and a channel size of $D$, five convolutional layers with a kernel size of three and a channel size of $2D$, and two convolutional layers with a kernel size of one and a channel size of $4D$. The output feature maps of the last convolutional layer in the encoder are projected into $D$-dimensional feature maps. They are utilized as the key for the encoder-decoder attentions. Likewise, the output feature maps of the last convolutional layer in the decoder are projected into $D$-dimensional feature maps for the final token classification.

**Training.**   In each iteration of the training phase, we fed each model a mini-batch containing approximately 10,000 tokens at most. In this setting, the batch size, or the number of the sentence pairs in a mini-batch, dynamically changes.

As a loss function, we utilized the cross entropy function with a label smoothing of 0.1.

**Optimization.**   We used the Riemannian Adam optimizer with $\beta_1 = 0.9$, $\beta_2 = 0.98$ and $\epsilon = 10^{-9}$ for both of the Euclidean and our hyperbolic architectures. For the scheduling of the learning rate $\eta$, we linearly increased the learning rate for the first 4000 iterations as a warm-up, and utilized the inverse square root decay with respect to the number of iterations $t$ thereafter as $\eta = (Dt)^{-\frac{1}{2}}$.

