# OpenReview forum: "Hyperbolic Neural Networks++"
_ICLR.cc/2021/Conference — ICLR 2021 Poster_

### Official Review · AnonReviewer1 · 2020-10-28
**How is HNN++ better than HNN in practice?**

**Rating:** 7
**Confidence:** 3

**Review:**

edit after rebuttal: The idea of the paper was interesting to me but some motivations or choices were unclear. After reading the rebuttal and other reviews, the authors have addressed most of my concerns. Therefore, I am ready to increase my score.

======

This paper introduces an improvement of the "Hyperbolic Neural Networks" (HNN) proposed by Ganea et al. and published at Neurips 2018.

The authors consider in particular the multinomial logistic regression problem. The probability score of a category is determined by a linear separator and a bias term.
In the vanilla HNN, two sets of parameters, each of dimensionality n (i.e. 2n in total), are learned to adapt the problem to hyperbolic geometry.
In the submission, the authors propose to learn only n+1 parameters per category. Thanks to this reformulation, the authors propose a new generalization of the fully connected layers of neural networks to hyperbolic geometry. This allows them to define arbitrary dimensional convolutional layers in the Poincare ball model. In Ganea et al., the authors do not exploit fully the hyperbolic geometry when they introduce their bias term, their discriminative hyperplane is instead determined in some tangent space (which is not hyperbolic), and then projected back to the hyperbolic space via the logarithm map. HNN++ proposes instead the exploit the hyperbolic geometry of their discriminator.

Although HNN and HNN++ are both hyperbolic neural networks, I think that the proposed approach should not be called HNN++ as it seems to suggest that this is an incremental extension of HNN.
In general, I think that the paper is hard to read. This is probably due to the 8 page format that makes some parts of the paper very dense. In particular:
- The split operation is not motivated until the multi-head attention part (i.e. 2 subsections after it was defined).
- What does the notation B(n/2,1/2) mean at the end of the "Generalization criterion"?
- The paper contains a lot of equations and formulae, but it would benefit from having a general pipeline explaining how to code exactly the approach. It currently seems hard to reimplement the approach, even with Section E in the Appendix. Will the code be available?

Concerning contributions, I do not think that the reduction of numbers of parameters from 2n to n+1 is a novel contribution, I also do not think it causes problems if the training data is large enough. However, I understand how proposing an approach that better exploits hyperbolic geometry to implement fully connected layers can help. My main concern is that the experiments do not seem to show how the proposed approach improves performance compared to HNN in practice.

- Section 4.1:
I do not understand the point of the experiments in Section 4.1. The scores obtained by HNN and HNN++ look very similar (are the scores on the right standard errors or standard deviations btw?).
It is mentioned in the last sentence of Section 4.1 that the "parameter-reduced approach obtains the same level of performance as a conventional hyperbolicMLR in a more stable training." What do the authors mean by more stable? Can they provide more details about the stability? Do they use the same optimizer between HNNs and the proposed approach?

- Why can't HNN be used as a baseline in Sections 4.2 and 4.3? Why is ConvSeq2Seq the only baseline in Table 3?

- Can the authors provide an experimental analysis on the contribution 4 (arbitrary dimensional convolutional layers) in page 2? How can tuning the "arbitrary dimensions" of convolutional layers have impact on performance?

- I have a general question about the gyromidpoints. I understand that the centroid of the squared Lorentzian distance is a minimizer over a sum of squared Lorentzian distances, just like the Euclidean centroid (average vector) would minimize a sum over squared Euclidean distance. However, is this also the case for the other types of gyromidpoints? What is the optimization problem(s) that those gyromidpoints minimize?


In conclusion, I would like to see an application where HNN++ is much better than HNN.


Minor comment: There is a minor mistake in the "Riemannian geometry" paragraph of Section 2. It is mentioned that g_x is a symmetric positive definite matrix.
This statement is correct for the Beltrami-Klein model because it is the sum of 2 symmetric positive definite matrices. However, this is not correct for the hyperboloid model because g_x is a diagonal matrix (see Section A of Appendix) and it would then be positive definite if and only if all its diagonal elements were positive.

---

> ### Author Response · Authors · 2020-11-21
> **Response to Reviewer #1 (Part 1)**
>
> We would like to thank the reviewer for his valuable comments.
>
> We would first like to emphasize that one of the contributions of our paper is the proposition of the hyperbolic version of convolutional layers and attention mechanisms in the Poincaré ball model, neither of which has been defined or proposed in HNNs. We now address all your questions and comments as follows:
>
> > The split operation is not motivated until the multi-head attention part (i.e., 2 subsections after it was defined).
>
> Although we have presented the specific examples of split and concatenation for convolutional layers and attention mechanisms in our paper, they can also be used in various other modules and situations. For this reason, we have not described the concrete motivation but stated a general motivation for their use in Section 3.3.
>
> > What does the notation B(n/2,1/2) mean at the end of the "Generalization criterion"?
>
> We apologize for the lack of a detailed description of the function B. B indicates a beta function.
>
> > The paper contains a lot of equations and formulae...Will the code be available?
>
> Because the basic architectures of the implemented models are the same as those of the Euclidean version, we do not think that there is much difficulty in that regard. As for the operations in the Poincaré ball model, they can be implemented by following and naturally extending the Geoopt library. Due to licensing concerns with Geoopt and Fairseq libraries, we are not ready to release the associated code. As soon as these issues are cleared, we intend to make a public release.
>
> > Concerning contributions, I do not think that the reduction of numbers of parameters from 2n to n+1 is a novel contribution, I also do not think it causes problems if the training data is large enough.
>
> Even though our method reduces the number of parameters, it is true that the parameter sizes of 2n and n+1 are both linear orders, so the amount of computation would not be dramatically reduced. On the other hand, when one redefines the FC layer using the MLR layer and extends the FC layer as a component of the convolutional layer and attention mechanism as in our proposed methods, the number of parameters in the entire network using such components will also be doubled if the number of parameters in the MLR layer remains twice as high. In this paper, our motivation was to avoid this by first reducing the number of parameters appropriately so that the number of parameters in the model does not change from that of the Euclidean counterpart when creating a network with any of the proposed components. In other words, halving the parameters of the hyperbolic MLR enables more viable construction of the neural networks in the hyperbolic space, which we consider is a worthy improvement compared with HNNs.
>
> > My main concern is that the experiments do not seem to show how the proposed approach improves performance compared to HNN in practice.
>
> For the MLR defined in Section 3.1, our goal was to achieve an extension keeping the compatible representational power with the MLR in HNNs while reducing the number of parameters to the halves. Table 1 demonstrates that the proposed MLR obtained the comparable accuracy using a smaller number of parameters, which we consider is the preferable aspect of our method.
>
> > Section 4.1: I do not understand the point of the experiments in Section 4.1. The scores obtained by HNN and HNN++ look very similar (are the scores on the right standard errors or standard deviations btw?). It is mentioned in the last sentence of Section 4.1 that the "parameter-reduced approach obtains the same level of performance as a conventional hyperbolic MLR in a more stable training." What do the authors mean by more stable? Can they provide more details about the stability? Do they use the same optimizer between HNNs and the proposed approach?
>
> We agree that the choice of word was not the best one. We would like to explain better what was meant.
>
> For the MLR defined in Section 3.1, our goal was to achieve an extension keeping the compatible representational power with the MLR in HNNs while reducing the number of parameters. This has been confirmed by the experiment in Section 4.1. (ibid)
>
> The experimental results show that F1 scores of the MLR in HNNs has much variability (i.e., some of the results are not good at all depending on the initial values of the parameters), whereas our proposed method was within the confidence interval of 1 order of magnitude percent, which confirms the stability of our method (in the sense of how the training changes when changing things as the seed or the initial parameters).
>
> As for the optimizer, as mentioned in the first paragraph of Section 4.1, we used the Riemannian Adam for all the experimented models, and the learning rate and the batch sizes were also aligned.

---

> > ### Author Response · Authors · 2020-11-21
> > **Response to Reviewer #1 (Part 2)**
> >
> > > Why can't HNN be used as a baseline in Sections 4.2 and 4.3? Why is ConvSeq2Seq the only baseline in Table 3?
> >
> > We agree that the explanation in the paper was not too clear and would like to explain it better.
> > As mentioned at the first of this response, neither a convolutional layer nor attention mechanism has been defined or proposed in HNNs, so we have not directly compared HNNs and HNNs++ for those components. To this reason, in Sections 4.2 and 4.3, we considered what kind of results can be obtained when existing methods in Euclidean space are turned into the hyperbolic space using our proposed methods.
> >
> > Table 2 shows that the models whose attention mechanisms in the Euclidean space were replaced by those of hyperbolic version with our methods were better suited for data distributed in the hyperbolic space, while maintaining comparable performance in the Euclidean space. We also confirmed that our methods can be trained in a more stable way without using any normalization layer, which is inevitably used in Euclidean neural networks deeper than a certain level.
> >
> > Table 3 shows that building a convolutional sequence-to-sequence model with operations in hyperbolic space using our proposed components significantly improves the representational capacity for natural language in low-dimensional spaces. We also discussed the prospects for improving performance in higher-dimensional spaces by taking advantage of this property.
> >
> > > Can the authors provide an experimental analysis on the contribution 4 (arbitrary dimensional convolutional layers) in page 2? How can tuning the "arbitrary dimensions" of convolutional layers have impact on performance?
> >
> > We showed in Section 3.4 that it is theoretically possible to construct an arbitrary dimensional convolutional layer, and in Section 4.3 we performed experiments with 1d convolutional layers as a suitable example utilizing natural language processing since it is one of the most relevant applications to hyperbolic spaces.
> >
> > > I have a general question about the gyromidpoints. I understand that the centroid of the squared Lorentzian distance is a minimizer over a sum of squared Lorentzian distances, just like the Euclidean centroid (average vector) would minimize a sum over squared Euclidean distance. However, is this also the case for the other types of gyromidpoints? What is the optimization problem(s) that those gyromidpoints minimize?
> >
> > A gyromidpoint is a geometric extension of the midpoint operation in the Euclidean space. In particular, it is a midpoint based on a distance metric called gyrometric defined in a gyrovector space (the norm between two gyrovectors$\|\|a {\ominus}_c b\|\|$).
> >
> > > Minor comment: There is a minor mistake in the "Riemannian geometry" paragraph of Section 2. It is mentioned that g_x is a symmetric positive definite matrix...However, this is not correct for the hyperboloid model.
> >
> > We appreciate your pointing it out. As you mentioned, the metric of the hyperboloid model ($g_\mathcal{L}$) is not positive definite as it is, but considering the "induced metric" corresponding to a specific parameterization of the manifold enables us to regard the hyperboloid model as a Riemannian manifold because it always becomes positive definite. Since this is not the most essential part in this paper, we have omitted the related explanation. We sincerely apologize for any confusion it may have caused.
> >
> > [1] Abraham Albert Ungar. A Gyrovector Space Approach to Hyperbolic Geometry. Synthesis Lectures
> > on Mathematics and Statistics, 1(1):1–194, 2009.
> >
> > [2] Models of Hyperbolic Plane, Willie WY Wong, 2019-09-16, (https://qnlw.info/post/models-of-hyperbolic-plane-201909/)

---

> > > ### Comment · AnonReviewer1 · 2020-11-23
> > > **Thank you for your response**
> > >
> > > I appreciate the response of the authors, I will update my score accordingly.
> > >
> > > Nonetheless, my question about gyromidpoints concerns a general set of points (more than two, as done in the paper). The authors only replied for the case of two points a and b.
> > >
> > > I still disagree with the current definition of the matrix g_x, it should be defined as a matrix such that v^\top g_x v > 0 for any non-vanishing tangent vector v.

---

> > > > ### Author Response · Authors · 2020-11-24
> > > > **Response to Reviewer #1**
> > > >
> > > > Thank you for your response. We greatly appreciate your questions and concerns towards the mathematical interpretation and correctness as we believe they would improve our paper and make it easer to understand.
> > > >
> > > > We now address all your additional comments as follows:
> > > >
> > > > > my question about gyromidpoints concerns a general set of points (more than two, as done in the paper). The authors only replied for the case of two points a and b.
> > > >
> > > > We sincerely apologize that we could not provide a sufficient answer to your concern in the previous response.
> > > > To the best of our knowledge, we could not find any prior work that directly mentions the relationship between the gyromidpoint and minimization problem of some function. Therefore, we instead proved the additional theorem about the properties of the Möbius gyromidpoint as a solution of a specific minimization problem:
> > > >
> > > > The Möbius gyromidpoint is a solution of the minimization problem of the weighted sum of calibrated squared gyrometrics, which is expressed as follows:
> > > > $\bar{b}=argmin_{\tilde{b}}\sum^N_{i=1}\nu_i\lambda^c_{\ominus_c\tilde{b}\oplus_c b_i}\|\ominus_c\tilde{b}\oplus_c b_i\|^2$.
> > > >
> > > > For more detail, please refer to Appendix D.7, which we have added in the latest revision.
> > > >
> > > >
> > > > > I still disagree with the current definition of the matrix g_x, it should be defined as a matrix such that v^\top g_x v > 0 for any non-vanishing tangent vector v.
> > > >
> > > > We sincerely apologize for any confusion it may have caused. We finally concluded that the description about the hyperboloid model in Appendix A was partially wrong, i.e., the matrix $g_\mathcal{L}$ cannot be the metric tensor of hyperboloid model, as you have pointed out. The clear reason is that $g_\mathcal{L}$ is an (n+1)-dimensional matrix whilst the manifold of the hyperboloid model is n-dimensional. We overlooked the fact that a metric tensor of an n-dimensional manifold as a sub-space of an (n+1)-dimensional space cannot be chosen from (n+1)-dimensional matrices and instead is formulated by another representation in terms of differential geometry, which caused these inaccuracies. We would like to emphasize again that, to indicate a metric tensor of the hyperboloid model as a matrix, we must give an n-dimensional coordinates, e.g., a hyperbolic polar coordinates.
> > > >
> > > > We corrected the expression in Appendix A according to the above reasons. We sincerely appreciate your letting us realize that we made a mathematical mistake and giving a chance to improve it.
> > > >
> > > > Including these two revised points, we uploaded a new revision, in which the changed parts are indicated as blue lines. We would like to thank you again for your constructive comments and would appreciate it if this revision could be any help.

---

### Official Review · AnonReviewer4 · 2020-10-28
**Hyperbolic Neural Networks++**

**Rating:** 6
**Confidence:** 4

**Review:**

This paper bolsters the library of hyperbolic neural network building blocks by defining a variety of layers in a natural way. These building blocks include a tighter parameterization of the basic fully connected layer, and the concatenate/split operations which are then used to define a version of hyperbolic multi-head attention.

The experimental section is adequate, covering standard hyperbolic applications such as embedding word hierarchies, as well as interesting tasks to validate the attention and convolutional capabilities of hyperbolic neural networks. Some details could be improved; for example, the BLEU score is difficult to interpret, the architecture and task details in Section 4.2 are sparse, and the first two experiments report only in very low dimensions which has been a toy testbed for hyperbolic models but can be unrealistic. Overall, the model shows general improvement over the baseline hyperbolic neural network models.

Overall, this paper does not present fundamentally new ideas in the development of non-Euclidean geometry for deep learning, but offers solid improvements in various aspects of constructing architecture components for hyperbolic neural networks and is a useful contribution.

---

> ### Author Response · Authors · 2020-11-21
> **Response to Reviewer #4**
>
> We would like to thank the reviewer for his valuable comments.
> We now address all your questions and comments as follows:
>
> > Some details could be improved; for example, the BLEU score is difficult to interpret, the architecture and task details in Section 4.2 are sparse, and the first two experiments report only in very low dimensions which has been a toy testbed for hyperbolic models but can be unrealistic.
>
> We apologize for the lack of a detailed description of the architecture and tasks in Section 4.2. As stated in the paper, we have made the experimental settings equivalent to the original paper and the published implementation of Set Transformer, so we think it would be sufficient to refer to them. We would like to explain it better in the final manuscript as we believe it would make the paper easier to understand.
>
> The reason why the small dimensional settings were exploited in the experiments 4.1 and 4.2 is because the original experiments of HNNs and Set Transformer, which were used for comparison, had the same settings (2, 3, 5, and 10 dimensions for HNNs, and 2 dimensional points for Set Transformer).
>
> We appreciate the suggestion that the BLEU score in Section 4.3 may be difficult to interpret. We would first like to explain the reason for using it. We adopted BLEU because it is the most commonly used metric to measure the performance of natural language processing models, especially for text generation and translation tasks.
>
> We apologize that we could not provide any other convincing metric for evaluating our method in Section 4.3. If you have any specific opinions on what other alternative metrics are available, we would appreciate your letting us know as we believe it would improve the paper.

---

### Official Review · AnonReviewer3 · 2020-10-28
**Official Blind Review #3**

**Rating:** 7
**Confidence:** 4

**Review:**

_Summary_:
Hyperbolic Neural Networks++ extends the existing work of applying hyperbolic manifolds to neural networks. It proposes new ways to reparametrize hyperbolic multinomial logistic regression (MLR) layers to reduce the number of parameters, to generalise fully connected layers  as well as split and concat operations to be more flexible and less computationally expensive. Further they expand hyperbolic neural networks to convolutional layer and attention mechanisms. The evaluation include direct comparison to HNN, and clustering with Transformers and seq-2-seq modeling.

_Strengths_:
- Paper is well-written and easy to understand. I also appreciate a good overview over hyperbolic geometry and consistent notation.
- The work proposed extends existing architectures with convolutional layers and attention mechanism which is prevalent in many deep architectures nowadays and thus an important contribution.
- The evaluation included a range of different tasks and architectures.

_Weaknesses_:
- The evaluation did not convince me completely of its superiority to HNNs. This is also not really discussed.

_Overall assessment_: I do believe that the extensions and generalization to Hyperbolic Neural Networks are important contributions. Therefore, I am recommending an accept.

_Detailed comments and questions_:
- Reduction of number of parameters needed for hyperbolic MLR: You correctly point out that the current formulation of hyperbolic MLR requires double the number of parameters in comparison to the Euclidean counterpart. This is reduced in this work. However, w.r.t. space complexity both are linear (=O(N)). Has the reduction in number of parameters other implications?
- Concatenation ("heavy computational cost"): Can you quantify that?
- Comparisons to HNN (Table 1): In 1/3 of the results HNN is superior to your approach? Do you have an explanation for that?

_Post-rebuttal_: I do appreciate the authors addressing my comments and updating their paper. Furthermore, the authors also addresses a lot of concerns of my fellow reviewers. As this was a good paper to begin with, I am keeping my score of recommending an accept.

---

> ### Author Response · Authors · 2020-11-21
> **Response to Reviewer #3**
>
> We would like to thank the reviewer for his valuable comments.
> We now address all your questions and comments as follows:
>
> > The evaluation did not convince me completely of its superiority to HNNs. This is also not really discussed.
>
> Because one of the contributions of our paper is the proposition of hyperbolic version of convolutional layers and attention mechanisms in the Poincaré ball model, neither of which has been defined or proposed in HNNs, only the MLR and FC layers are directly comparable between HNNs and HNNs++ in our paper.
>
> For the MLR defined in Section 3.1, our goal was to achieve an extension keeping the compatible representational power with the MLR in HNNs while reducing the number of parameters to the halves. Table 1 demonstrates that the proposed MLR obtained the comparable accuracy using a smaller number of parameters, which we consider is the preferable aspect of our method.
> We have not conducted independent experiments on the FC layer for the following reasons:
>
> -	The difficulty of creating an appropriate experimental setup that focuses only on FC layers and is related to the hyperbolic space. In particular, the lack of an experimental setup that can be used as a reference because no experimental evaluation of the FC layer has been done in HNNs.
>
> Instead, the performance of the FC layer was indirectly evaluated in Section 4.2 and 4.3 by incorporating it into convolutional layers and attention mechanisms.
>
> > Reduction of number of parameters needed for hyperbolic MLR: You correctly point out that the current formulation of hyperbolic MLR requires double the number of parameters in comparison to the Euclidean counterpart. This is reduced in this work. However, w.r.t. space complexity both are linear (=O(N)). Has the reduction in number of parameters other implications?
>
> Even though our method reduces the number of parameters, it is true that the parameter size of 2n and n+1 are both linear orders, so the amount of computation would not be dramatically reduced. On the other hand, when one redefines the FC layer using the MLR layer and extends the FC layer as a component of the convolutional layer and attention mechanism as in our proposed methods, the number of parameters in the entire network using such components will also be doubled if the number of parameters in the MLR layer remains twice as high. In this paper, our motivation was to avoid this by first reducing the number of parameters appropriately so that the number of parameters in the model does not change from that of the Euclidean counterpart when creating a network with any of the proposed components. In other words, halving the parameters of the hyperbolic MLR enables more viable construction of the neural networks in the hyperbolic space, which we consider is a worthy improvement compared with HNNs.
>
> > Concatenation ("heavy computational cost"): Can you quantify that?
>
> The concatenation proposed in HNNs requires multiple Möbius additions for the number of vectors to be added ($N$), but the Möbius addition is non-commutative and non-associative, so the input vectors cannot be added in parallel like normal vector addition. In other words, the addition must be repeated $N-1$ times sequentially. On the other hand, our proposed concatenation requires only the operations of an exponential map and logarithmic map, regardless of $N$.
>
> > Comparisons to HNN (Table 1): In 1/3 of the results HNN is superior to your approach? Do you have an explanation for that?
>
> We agree that the explanation in the paper was not too clear and some of the results of HNNs were better scores than our method.
>
> We would like to emphasize that, for the MLR defined in Section 3.1, our goal was to achieve an extension keeping the compatible representational power with the MLR in HNNs while reducing the number of parameters to the halves. Table 1 demonstrates that the proposed MLR indeed obtained the comparable accuracy because for almost all of the results in which the average test scores of HNNs were higher than those of HNNs++, the confidence intervals overlap.

---

### Official Review · AnonReviewer2 · 2020-10-28
**Review of "Hyperbolic Neural Networks++"**

**Rating:** 8
**Confidence:** 4

**Review:**

Paper summary:

The paper provides a reformulation of the fundamental operations in Euclidean space that are used in neural networks for the Poincaré ball model of hyperbolic space (and thus hyperbolic space generally). The paper’s reformulation differs from previous reformulations (Ganea et al. 2018) in several ways. For multinomial logistic regression, the excess n-1 degrees of freedom available in previous formulations when defining a Poincaré hyperplane via a point on the hyperplane and a normal vector are eliminated by using a canonical choice of normal vector along with a scalar quantity corresponding to the distance to the hyperplane from the origin. Fully connected (FC) neural network layers are also reformulated in a way that keeps with the interpretation of an affine transformation as returning a point whose individual coordinates are distances to a set of different hyperplanes. On the other hand, the previous reformulation directly used Möbius matrix-vector multiplication, which does not this same interpretation. The paper then gives hyperbolic reformulations of further types of neural network operations in Euclidean space, namely split/concatenation (less computationally intensive than that in previous work), convolution (not present in previous work). Turning its focus to attention models, the paper proves a theorem regarding the equivalence of various hyperbolic midpoints proposed in previous work. Finally, the paper carries out experiments testing each part of its reformulation on appropriate datasets.

------------------------------------------
Strengths and weaknesses:

The paper gave clear motivations for each of part of its hyperbolic reformulation and explained how it differed from previous reformulations. The paper was well-written, and the authors performed regular sanity checks, such as re-deriving the Euclidean case for c -> 0. The experiments were appropriate and demonstrated that the reformulation works as well as previous reformulations (possibly better, with regard to stability). I liked the paper a lot, and I think it will definitely be of interest to people working on non-Euclidean deep learning. I’m assigning a score of 8 (a very good conference paper), and I think that the paper is more or less ready for publication as is. I’ve included a few questions below (to help my own understanding), as well as some typos I spotted whilst reading the paper.

------------------------------------------
Questions and clarification requests:

1) Which of Spivak (1979), Petersen et al. (2006), and Andrews & Hopper (2010) were you working from when writing the brief summary of Riemannian Geometry on page 2? Some of the definitions given like “An n-dimensional manifold M is an n-dimensional topological space that can be linearly approximated to an n-dimensional real space at any point x ∈ M” and “g_{x} is a positive definite symmetric matrix defined on T_{x}M” struck me as unusual (and possibly imprecise).
2) In equation 4, you write q as the exponential of r_{k}[a_{k}], even though a_{k} ∈ T_{q}M, not T_{0}M. Is this correct?
3) At the beginning of section 3, you write “The core concept is re-generalization of ⟨a,x⟩−b type equations with no increase in the number of parameters, which has the potential to replace any affine transformation in a shared manner.” What did you mean by “in a shared manner”?
4) At the end of section 4.1, you write “In particular, our parameter-reduced approach obtains the same level of performance as a conventional hyperbolic MLR in a more stable training, as can be seen from the relatively narrower confidence intervals.” Do you think your approach is genuinely more stable? If so, do you have any intuition as to why performance was more stable?

------------------------------------------
Typos and minor edits:
- Section 1, paragraph 4, sentence 1 – “Despite such progresses” -> “Despite such progress”
- Section 2, Poincaré hyperplane – Ganea et al. 2018 certainly discusses Poincaré hyperplanes, but hyperplanes in Riemannian geometry have been studied a very long time before this.
- Section 3.3, paragraph 1, sentence 5 – “approaches to the” -> “approaches the”
- Section 3.3, last paragraph, last sentence – “treats every inputs fairly” -> “treats every input fairly”
- Section 4.2, paragraph 1, sentence 2 – better to rephrase this
- Section 4.2, paragraph 2, sentence 2 – “our models achieved the almost the same performance as Set Transformers” -> “our models achieved almost the same performance as Set Transformers”
- Section 4.3, paragraph 1, sentence 1 – “we experimented the convolutional sequence to sequence modelling task” -> “we experimented with the convolutional sequence-to-sequence modelling task”
- Table 2 – “on the Euclidean space” -> “on Euclidean space”
- Section 4.3, paragraph 1, sentence 2 – “the hyperbolic version of which have already been verified” -> “the hyperbolic version of which has already been verified”
- Section 4.3, paragraph 1, sentence 6 – “Note that the inputs for the sigmoid functions in Gated Linear Units are logarithmic mapped just as hyperbolic Gated Recurrent Units proposed by” -> “Note that the inputs for the sigmoid functions in Gated Linear Units are logarithmically mapped just like hyperbolic Gated Recurrent Units proposed by”
- Conclusion, final sentence – “for the future researches” -> “for future research” / “for future researchers”

---

> ### Author Response · Authors · 2020-11-21
> **Response to Reviewer #2**
>
> We would like to thank the reviewer for his valuable comments. We would also like to express our gratitude towards you for letting us know our mistakes and misspellings in the manuscript. We uploaded the revised version reflecting your corrections.
> We now address all your questions and comments as follows:
>
> > Which of Spivak (1979), Petersen et al. (2006), and Andrews & Hopper (2010) were you working from when writing the brief summary of Riemannian Geometry on page 2? Some of the definitions given like “An n-dimensional manifold M is an n-dimensional topological space that can be linearly approximated to an n-dimensional real space at any point x ∈ M” and “g_{x} is a positive definite symmetric matrix defined on T_{x}M” struck me as unusual (and possibly imprecise).
>
> We sincerely apologize for any confusion it may have caused. For a more complete and thorough explanation about the topic of Riemannian geometry we refer the reader to Spivak (1979), Petersen et al. (2006), and Andrews & Hopper (2010) and not directly quoted them to the contents in our paper. For the explanation in Section 2, we simplified our mathematical knowledge for the purpose of introducing the notations and letting readers capture the overview needed for the rest of the paper, which might have resulted in some inaccuracies. We would appreciate it if you could tell us how you think the parts you have pointed out should be specifically corrected. We would like to make these corrections as we believe they would make the paper easier to understand.
>
> > In equation 4, you write q as the exponential of r_{k}[a_{k}], even though a_{k} ∈ T_{q}M, not T_{0}M. Is this correct?
>
> Thank you for the insightful question.
>
> Because $q$ in Eq. 4 corresponds to $p$ in HNNs (and also mentioned in Section 3.1), this is a bias vector rooted at the origin. Also, although there is a term of $a_k$ in $exp^c_0$, we no longer need to consider where $a_k$ is a tangent vector rooted at, because it is normalized by the [・] operation, i.e., only its direction is taken into account. This is also due to the property of the conformal nature of the Poincaré ball model. Therefore, $q$ can be regarded as the exponential map of $r_{k}[a_{k}]$ at the origin.
>
> > At the beginning of section 3, you write “The core concept is re-generalization of ⟨a,x⟩−b type equations with no increase in the number of parameters, which has the potential to replace any affine transformation in a shared manner.” What did you mean by “in a shared manner”?
>
> We agree that the explanation in the paper was not too clear. We would like to explain better what was meant.
>
> In the conventional HNNs, the MLR and the FC layer were formulated based on different mathematical insights. On the other hand, our proposed method defines the FC layer based on the pre-reformulated MLR, and furthermore uses it to define the following convolutional layers and attention mechanisms. In that sense, we described such a straight relation as "in a shared manner".
>
> We would like to explain it better in the final manuscript. We consider that “based on the same mathematical principle” would serves the better alternative, but if there is any suggestion or another comment for it, we would really appreciate it.
>
> > At the end of section 4.1, you write “In particular, our parameter-reduced approach obtains the same level of performance as a conventional hyperbolic MLR in a more stable training, as can be seen from the relatively narrower confidence intervals.” Do you think your approach is genuinely more stable? If so, do you have any intuition as to why performance was more stable?
>
> The experimental results show that F1 scores of the MLR in HNNs has much variability (i.e., some of the results are not good at all depending on the initial values of the parameters), whereas our proposed method was within the confidence interval of 1 order of magnitude percent, which confirms the stability of our method (in the sense of how the training changes when changing things as the seed or the initial parameters). Although we have not derived a theoretical bound of the stability, we believe that this is due to the fact that reducing the parameters eliminates the redundant degrees of freedom, making it easier to learn and less likely to fall into local minima.

---

### Author Response · Authors · 2020-11-25
**General notes and summary of revisions**

We wish to thank all four reviewers for their helpful comments and feedbacks.
Based on the individual responses, we have uploaded a revised version of the manuscript, in which the changed parts are indicated as blue lines.

The summary of the revision is as follows.

- We corrected some grammatical mistakes and misspellings.
- We added some explanations related to our proposed methods, experimental settings, and experimental results in Sections 2, 3.1, 3.3, 3.5, 4.1, and 4.2 to make the manuscript better and clearer.
- We changed the words "cosine similarity" in Section 3.5 to the correct notation "inner product" (This correction would not cause any additional changes in the contents of the paper because we made a mistake only for the expression.)
- We corrected the notation of the hyperboloid model in Appendix A about its metric tensor, and wrote an additional explanation.
- We proved additional theorem about the Möbius gyromidpoint in Appendix D.7.

We would like again to express our gratitude towards all the reviewers for their constructive comments and would appreciate it if this revision could be any help.

---

### Decision · Program_Chairs · 2021-01-07
**Final Decision**

**Decision:**

Accept (Poster)

**Comment:**

The paper introduces new methods and building blocks to improve hyperbolic neural networks, including a tighter parameterization of fully connected layers, convolution, and concatenate/split operations to define a version of hyperbolic multi-head attention. The paper is well written and relevant to the ICLR community. The proposed methods offer solid improvements over previous approaches in various aspects of constructing hyperbolic neural networks and also extends their applicability. As such, the paper provides valuable contributions to advance research in learning non-Euclidean representations and HNNs. All reviewers and the AC support acceptance for the paper's contributions. Please consider revising your paper to take feedback from reviewers after reubttal into account.